# Concerted regulation of ISWI by an autoinhibitory domain and the H4 N-terminal tail

Johanna Ludwigsen[1], Sabrina Pfennig[1], Ashish K Singh[1], Christina Schindler[2,3], Nadine Harrer[1], Ignasi Forné[1], Martin Zacharias[2,3], Felix Mueller-Planitz[1]*

[1]Biomedical Center, Ludwig-Maximilians-Universität München, Munich, Germany; [2]Physics Department (T38), Technische Universität München, Munich, Germany; [3]Center for Integrated Protein Science Munich, Munich, Germany

**Abstract** ISWI-family nucleosome remodeling enzymes need the histone H4 N-terminal tail to mobilize nucleosomes. Here we mapped the H4-tail binding pocket of ISWI. Surprisingly the binding site was adjacent to but not overlapping with the docking site of an auto-regulatory motif, AutoN, in the N-terminal region (NTR) of ISWI, indicating that AutoN does not act as a simple pseudosubstrate as suggested previously. Rather, AutoN cooperated with a hitherto uncharacterized motif, termed AcidicN, to confer H4-tail sensitivity and discriminate between DNA and nucleosomes. A third motif in the NTR, ppHSA, was functionally required in vivo and provided structural stability by clamping the NTR to Lobe 2 of the ATPase domain. This configuration is reminiscent of Chd1 even though Chd1 contains an unrelated NTR. Our results shed light on the intricate structural and functional regulation of ISWI by the NTR and uncover surprising parallels with Chd1.

*For correspondence: felix. mueller-planitz@med.uni-muenchen.de

**Competing interests:** The authors declare that no competing interests exist.

## Introduction

Eukaryotic cells package their DNA into chromatin. Chromatin organization allows cells to compact, protect and regulate their genomes. Nucleosomes are the primary building blocks of chromatin. These particles consist of ~150 bp of DNA that wrap almost twice around an octamer of histones. Nucleosomal DNA, however, is not accessible to most nuclear factors. Nature therefore evolved ATP-dependent nucleosome remodeling complexes that can alter the position or the structure of nucleosomes as necessary.

Numerous remodeling complexes with distinct activities are active in any cell. Some move nucleosomes along DNA, eject histones or exchange them for histone variants, and some can even perform several of these activities (*Zhou et al., 2016*). How the various remodelers are regulated in response to cellular needs is not well understood. Several remodelers, for instance, respond to post-translational modifications present on histones (*Swygert and Peterson, 2014*). Others are directly regulated by post-translational modifications (*Kim et al., 2010*) or react to small signaling molecules (*Zhao et al., 1998*). Cells also adjust the subunit composition of remodeling complexes during development (*Lessard et al., 2007*). All these examples indicate exquisite levels of controls exerted over remodeling complexes. The fact that mutations in subunits of human remodeling factors strongly associate with and in some cases drive cancers underscores the necessity to regulate remodeler activity (*Kadoch and Crabtree, 2015*; *Garraway and Lander, 2013*).

Remodelers of the ISWI family – like most other remodelers – can reposition nucleosomes along DNA in a process termed nucleosome sliding. ISWI's activity is directly regulated by the histone H4 N-terminal tail and by DNA that flanks the nucleosome, so called linker DNA. The regulation

**eLife digest** In the cells of animals, plants and other eukaryotes, DNA wraps tightly around proteins called histones to form structures known as nucleosomes that resemble beads on a string. When nucleosomes are sufficiently close to each other they interact and clump together, which compacts the DNA and prevents the genes in that stretch of DNA being activated.

But how do cells mobilize their nucleosomes? A nucleosome remodeling enzyme called ISWI can slide nucleosomes along DNA. ISWI becomes active when it interacts with a 'tail' region of a histone protein called H4. However, the H4 tail prefers to interact with neighboring nucleosomes instead of with ISWI. Therefore when ISWI slides a nucleosome close to another one, the H4 tail of the nucleosome binds instead to its new neighbor so that ISWI cannot continue to slide. By this mechanism, ISWI is proposed to pile up nucleosomes, which subsequently compact, leading to the inactivation of this part of the genome.

To investigate how ISWI recognizes the H4 tail, Ludwigsen et al. mapped where the H4 tail binds to ISWI by combining the biochemical methods of cross-linking and mass spectrometry. In addition, mutagenesis experiments identified a new motif in the enzyme that is essential for recognizing the H4 tail. In the absence of the nucleosome, this motif – called AcidicN – works with a neighboring motif called AutoN to keep ISWI in an inactive state. The two motifs also work together to enable ISWI to distinguish between nucleosomes and DNA. Further evidence suggests that other remodeling enzymes have similar regulation mechanisms; therefore this method of controlling nucleosome remodeling may have been conserved throughout evolution.

Further studies are now needed to detect the shape changes that occur in ISWI as it recognizes the histone tail and work out how this leads to nucleosome remodeling. Inside cells, ISWI is usually found within large complexes that consist of many proteins. It therefore also remains to be discovered whether the proteins in these complexes impose additional layers of regulation and complexity on the activity of ISWI.

imposed by these epitopes has direct consequences for the biological output of ISWI remodelers. By measuring the length of linker DNA, ISWI can generate arrays of evenly spaced nucleosomes (*Lieleg et al., 2015*; *Yang et al., 2006*; *Yamada et al., 2011*), a characteristic feature of chromatin. Arrays of nucleosomes can further compact. In the compacted state, the histone H4 N-terminal tail of one nucleosome contacts the acidic patch formed by H2A and H2B of a neighboring nucleosome (*Luger et al., 1997*; *Dorigo et al., 2004*). This interaction sequesters the H4 tail, which now is no longer available for binding to and stimulating the activity of ISWI. Thus, ISWI's activity on the compacted chromatin would decrease, ensuring the unidirectionality of the reaction. This process is in line with the importance of some ISWI complexes in heterochromatin biology (*Bozhenok et al., 2002*).

How ISWI senses the H4 tail is largely unknown. Evidence points to the ATPase domain of ISWI directly binding the H4 tail (*Racki et al., 2014*; *Mueller-Planitz et al., 2013*), consistent with the tail directly influencing catalytic reaction steps (*Clapier et al., 2001*; *Dang et al., 2006*). However, a domain at the C-terminal side of ISWI, the HAND-SANT-SLIDE (HSS) domain, has been implicated in binding the H4 tail as well (*Boyer et al., 2004*; *Grüne et al., 2003*). Another layer of regulation is imposed by the non-catalytic subunit termed ACF1, which associates with ISWI and sequesters the H4 tail under certain conditions (*Hwang et al., 2014*).

ISWI recognizes amino acids $R_{17}H_{18}R_{19}$ within the H4 tail, which are part of a stretch of amino acids called basic patch (*Fazzio et al., 2005*; *Hamiche et al., 2001*; *Clapier et al., 2002*). Notably, ISWI contains an identical motif, here called AutoN. Mutation of AutoN's two arginines to alanines (referred to as 2RA) increased the DNA-stimulated ATPase activity and nucleosome sliding, and suppressed the dependence of ISWI's ATPase and sliding activities on the H4 tail. According to the current model, AutoN directly binds to and blocks the H4-tail binding site, acting as a gatekeeper for the H4 tail. This model necessitates a conformational change of the NTR to allow binding of H4 (*Hwang et al., 2014*; *Clapier and Cairns, 2012*). Indeed, a conformational change could be traced to AutoN upon nucleic acid binding (*Mueller-Planitz et al., 2013*). Of note, the 2RA mutation

diminished but did not abolish the H4-tail dependency, implicating also other regions in the H4 recognition process (*Clapier and Cairns, 2012*).

The AutoN motif is embedded in a structurally and functionally poorly characterized domain referred to as the N-terminal region (NTR). Besides AutoN, the NTR contains additional motifs: an acidic region that we termed AcidicN, the 'post-post-helicase-SANT-associated' (ppHSA) motif, so named because it follows the post-HSA motif in remodelers of the Snf2 family (*Mueller-Planitz et al., 2013*; *Szerlong et al., 2008*), and a weakly conserved AT-hook (*Mueller-Planitz et al., 2013*; *Aravind and Landsman, 1998*). Their functions remain unknown.

Here, we systematically interrogated the functions of all conserved motifs within the NTR by mutagenesis and a series of quantitative biochemical assays in vitro and in vivo. We paid particular attention to probe for possible crosstalk between these motifs and the H4 tail to understand its recognition process. Using protein crosslinking followed by mass spectrometry and protein structural modeling we obtained information about the general structural architecture of the NTR-ATPase module. With similar approaches, we mapped the H4-tail binding site. We interpret our results within a unified structural and functional framework for the combined inhibition of ISWI by the NTR and recognition of the histone H4 tail. Contrary to current models, we propose that AutoN does not occlude the binding pocket of the H4 tail and that inhibition by AutoN involves a more elaborate mechanism than simple mimicry of the H4 basic patch.

## Results

### The NTR contains conserved motifs

Multiple sequence alignment of ISWI homologs revealed several sequence motifs in the NTR of ISWI (*Mueller-Planitz et al., 2013*). To assess their degree of conservation we queried the UniProt database for ISWI homologs (*Figure 1—figure supplement 1*). Sequence alignment of these candidates showed conservation of AutoN (*Clapier and Cairns, 2012*) but also indicated that two other motifs, termed ppHSA and AcidicN, were at least as conserved (*Figure 1*). In contrast, an AT-hook (*Aravind and Landsman, 1998*) was poorly conserved. Of note, a separate PSI-BLAST of the NTR of ISWI revealed conservation of ppHSA across multiple families of remodelers, including Snf2, Lsh and Ino80, suggesting shared function (*Figure 1F*). ppHSA and AcidicN have not been characterized so far.

### The ppHSA motif is important for structural stability

To study its physiological role, we serially truncated the NTR of Isw1 in *Saccharomyces cerevisiae* (*Figure 2A*) and tested whether these truncation variants complemented a previously characterized growth defect of a yeast triple knockout (TKO) strain lacking three remodelers (*ΔISW1, ΔISW2, ΔCHD1*) at elevated temperatures (*Tsukiyama et al., 1999*). To assess whether complementation was dependent on the expression level, the alleles were placed under the control of synthetic promoters of varying strengths (*Blazeck et al., 2012*). Protein expression levels were measured by Western blot analysis (*Figure 2—figure supplement 1E*).

Expression of none of the N-terminal truncation variants fully complemented the growth phenotype, indicating functional relevance of the NTR in vivo. In contrast, the TKO strain that was complemented with full-length Isw1 grew essentially as well as the *ΔISW2, ΔChd1* double knockout strain (DKO; *Figure 2B*, *Figure 2—figure supplement 1A*). Isw1 variants that lacked the AutoN-AcidicN region in addition to ppHSA grew modestly better than Isw1$_{ΔppHSA}$, in line with the general inhibitory nature of AcidicN and AutoN (compare rows 1 and 2 of *Figure 2—figure supplement 1B,C* to the same rows in D; see also below).

We noted a pronounced toxicity of all Isw1 mutants as indicated by slow growth at elevated expression levels (for instance, compare row four with row five in *Figure 2—figure supplement 1B, C,D*). Full-length Isw1, on the other hand, was not toxic at comparable expression levels (*Figure 2—figure supplement 1A*).

Toxicity at high expression levels could be caused by structural instability of the N-terminally truncated Isw1 variants. Indeed, analogous ISWI derivatives from *Drosophila melanogaster* proved difficult to purify (see below), supporting the notion that mutations in the NTR destabilize ISWI structure.

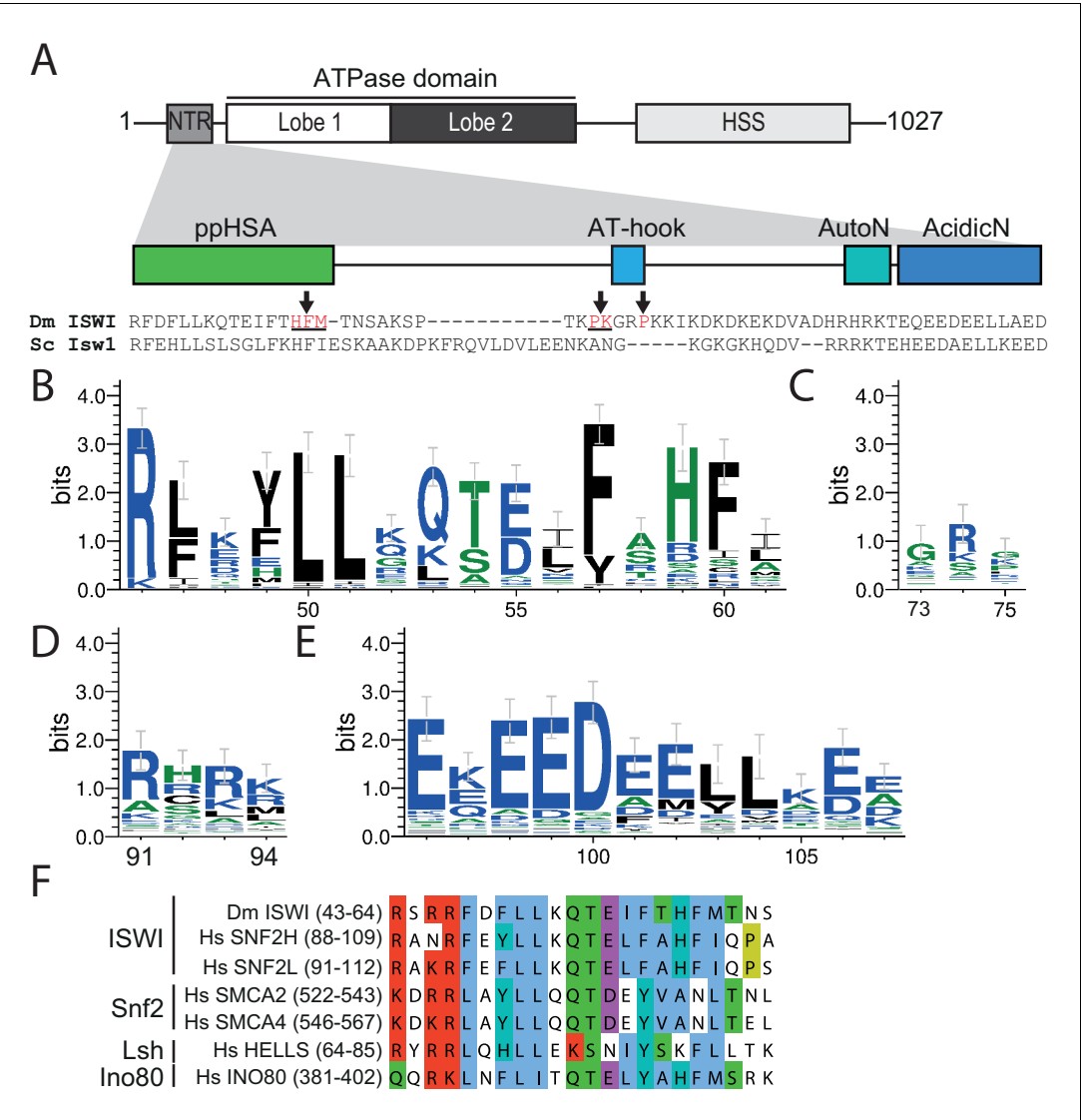

**Figure 1.** The NTR of ISWI contains several conserved sequence motifs. (**A**) Schematic representation of the ISWI domain composition. The grey inset shows the sequence and motifs of the NTR. Arrows indicate amino acids within the NTR of *Drosophila* ISWI that crosslinked to Lobe 2 of the ATPase domain (*Table 1*). HSS, HAND-SANT-SLIDE domain. (**B**–**E**) Sequence logos showing the sequence conservation of (**B**) ppHSA, (**C**) AT-hook, (**D**) AutoN, and (**E**) AcidicN. X-Axis values are amino acid positions in *D. melanogaster* ISWI. See *Figure 1—figure supplement 1* for full alignment. (**F**) Alignment of the ppHSA motif of *Drosophila* (Dm) ISWI with the human (Hs) ISWI homologs SNF2H and SNF2L and representatives of unrelated remodeler families.

The following figure supplement is available for figure 1:

**Figure supplement 1.** Alignment of ISWI homologs from various organisms.

## The ppHSA motif does not substantially contribute to catalysis

Toxicity of the Isw1 NTR deletions precluded a detailed analysis in vivo. Importantly, the in vivo results left open the possibility that NTR-deleted Isw1 was catalytically inactive. We therefore continued to study the function of the NTR motifs in vitro using purified *Drosophila* ISWI proteins.

Although ISWI variants carrying mutations or deletions in the NTR generally expressed well, we failed to purify them using standard protocols. For each ISWI variant, we screened through a variety of expression and purification strategies to improve the yield of soluble protein. The strategies that

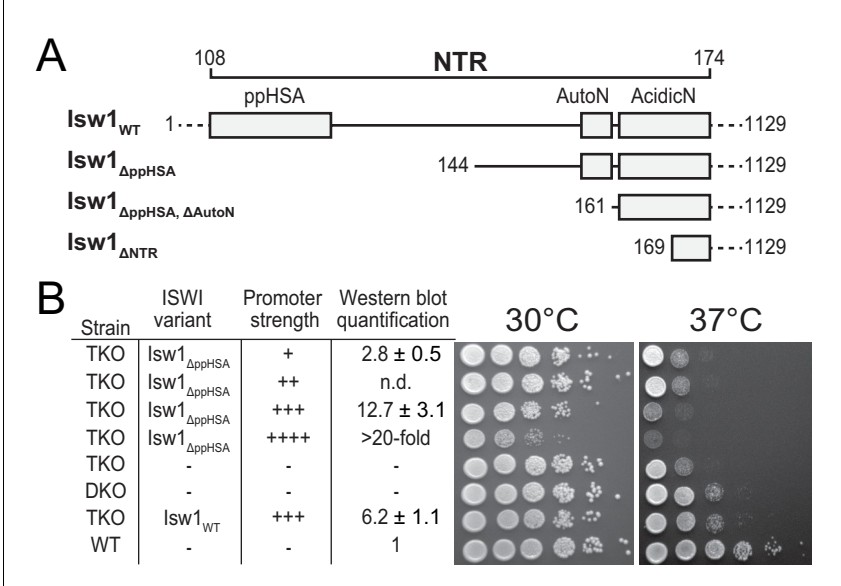

**Figure 2.** Functional importance of the NTR of yeast Isw1 in vivo. (**A**) Successive N-terminal truncation mutants of Isw1. Note that Isw1$_{ΔNTR}$ lacked the entire N-terminus up to the first seven residues of AcidicN (**Figure 1E**). (**B**) Complementation assay with Isw1$_{ΔppHSA}$. A yeast strain lacking *ISW1, ISW2 and CHD1* (TKO) was transformed with Isw1 derivatives under control of promoters of varying strengths. In comparison to a strain lacking only *ISW2* and *CHD1* (DKO), Isw1$_{WT}$ fully complemented the growth phenotype at elevated temperatures (37°C). In contrast, Isw1$_{ΔppHSA}$ did not complement at any expression level. Results for other Isw1 variants can be found in **Figure 2—figure supplement 1**. Growth was assessed by spotting tenfold serial dilutions of liquid cultures.

The following figure supplement is available for figure 2:

**Figure supplement 1.** Complementation assay with N-terminal truncation variants of Isw1.

we employed included fusion to solubility tags (Z$_2$, GB1, NusA, TrxA), fusion to or co-expression of chaperones (trigger factor, GroES/GroEL, DnaK/DnaJ/GrpE) and inclusion of protease sites (3C) at three locations in the NTR to cleave off parts of the N-terminus after purification. The strategies that proved successful are summarized schematically in **Figure 3—figure supplement 1** and **Figure 6—figure supplement 1**.

We first benchmarked the DNA- and chromatin-stimulated ATPase activities of ISWI that lacked ppHSA (ISWI$_{ΔppHSA}$) or both ppHSA and AT-hook (ISWI$_{ΔppHSA; ΔAT-hook}$) against the activity of wild-type ISWI (ISWI$_{WT}$). We used saturating ATP and nucleic acid concentrations as indicated by control experiments with varying levels of ligands (**Figure 3—figure supplement 2**). DNA- and chromatin-stimulated ATPase rates of the truncation mutants varied by no more than 1.8-fold from ISWI$_{WT}$ (**Figure 3A,B**) indicating that ppHSA and AT-hook were largely dispensable for ATP hydrolysis and for proper recognition of chromatin.

To evaluate whether ppHSA and AT-hook were required to efficiently couple ATP hydrolysis to nucleosome remodeling, we employed a quantitative remodeling assay. This assay monitors remodeling of a single nucleosome in the context of a 25-mer nucleosomal array by measuring the remodeling-dependent exposure of a unique restriction enzyme site originally occluded by the nucleosome (**Mueller-Planitz et al., 2013**). Time courses of the remodeling reaction were fit to single exponential functions to extract the observed remodeling rate constant $k_{obs}$ (**Figure 3C**; **Figure 3—figure supplement 3**), which provided us with a quantitative measure to compare the remodeling activities of ISWI and its derivatives.

Remodeling was affected only modestly by deletion of parts of the NTR (3.3- and 1.4-fold for ISWI$_{ΔppHSA}$ and ISWI$_{ΔppHSA, ΔAT-hook}$, respectively; **Figure 3C**). In conclusion, ATPase and remodeling data suggested that both ppHSA and AT-hook are not absolutely required for catalysis in vitro. The

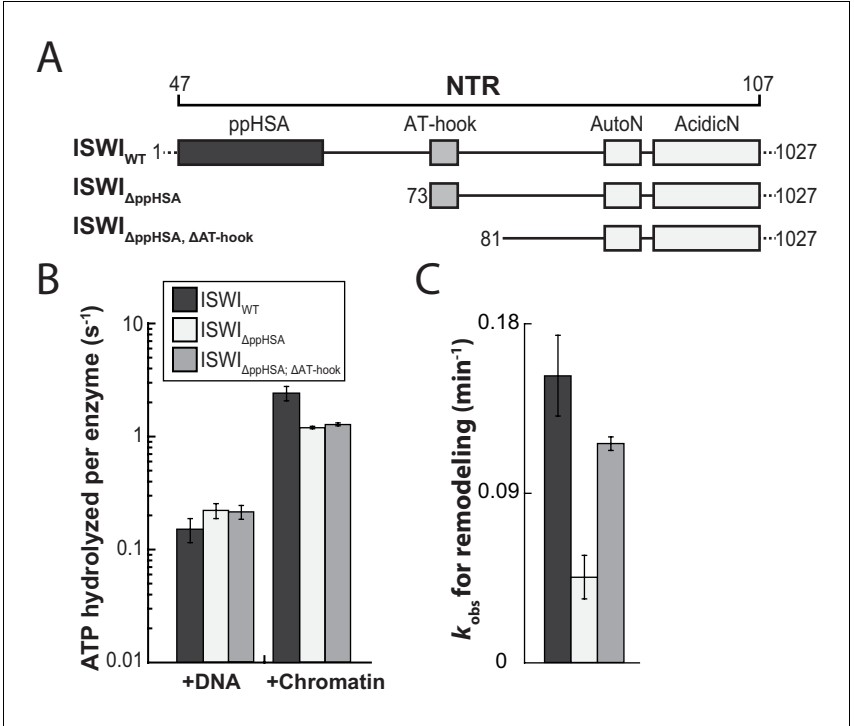

**Figure 3.** The ppHSA motif is largely dispensable for catalysis. (**A**) N-terminal truncation mutants of *Drosophila* ISWI. (**B**) DNA- and nucleosome-stimulated ATP turnover. ATPase rates were measured in the presence of saturating concentrations of ATP (1 mM), DNA (0.2 g/l) or nucleosomes (0.1 g/l). Errors for nucleosome-stimulated rates of ISWI deletion mutants are minimal and maximal values of two independent measurements, and s.d. for all other measurements (n ≥ 4). ATPase rates in absence of nucleic acids were <0.022 s$^{-1}$ for all ISWI variants (data not shown). (**C**) Remodeling activity was determined by measuring the accessibility changes of a unique KpnI restriction site in a 25-mer nucleosomal array (100 nM nucleosomes, 300 nM enzyme). Errors are s.d. (n ≥ 3) except for ISWI$_{\Delta ppHSA; \Delta AT\text{-}hook}$ for which minimal and maximal values of two independent measurements are shown. Raw data of the remodeling assay can be found in *Figure 3—figure supplement 3*. Color code as in panel **B**.

The following figure supplements are available for figure 3:

**Figure supplement 1.** Cloning and purification of N-terminal truncation variants of *Drosophila* ISWI.

**Figure supplement 2.** Saturation controls for ISWI$_{WT}$ and ISWI$_{\Delta ppHSA}$ in ATPase assays.

**Figure supplement 3.** Determination of the rate constants for remodeling ($k_{obs}$; *Figure 3C*) for ISWI$_{WT}$ and N-terminal truncation mutants of ISWI.

---

modest decreases in remodeling activities could be due to lower stability of these enzymes (see above).

## The NTR contacts Lobe 2 of the ATPase domain

We speculated that the NTR might stabilize the structure of ISWI by adopting a similar configuration as the two chromo domains of the related remodeler Chd1. Like the NTR, the chromo domains are located directly N-terminal to the ATPase module. Notably, they bridge over and pack against the second ATPase lobe, presumably locking the ATPase in an inactive state (*Figure 4A*) (*Hauk et al., 2010*).

To explore, we first determined the binding interface of the chromo domains (amino acids 239–284) on Lobe 2 of the ATPase module using the PISA algorithm (www.ebi.ac.uk/pdbe/pisa/) and visualized the analogous surface on a homology model of ISWI (*Figure 4B*; cyan). We then site-specifically inserted the UV-crosslinking amino acid *p*-benzoyl-*p*-phenylalanine (abbreviated Bpa or B)

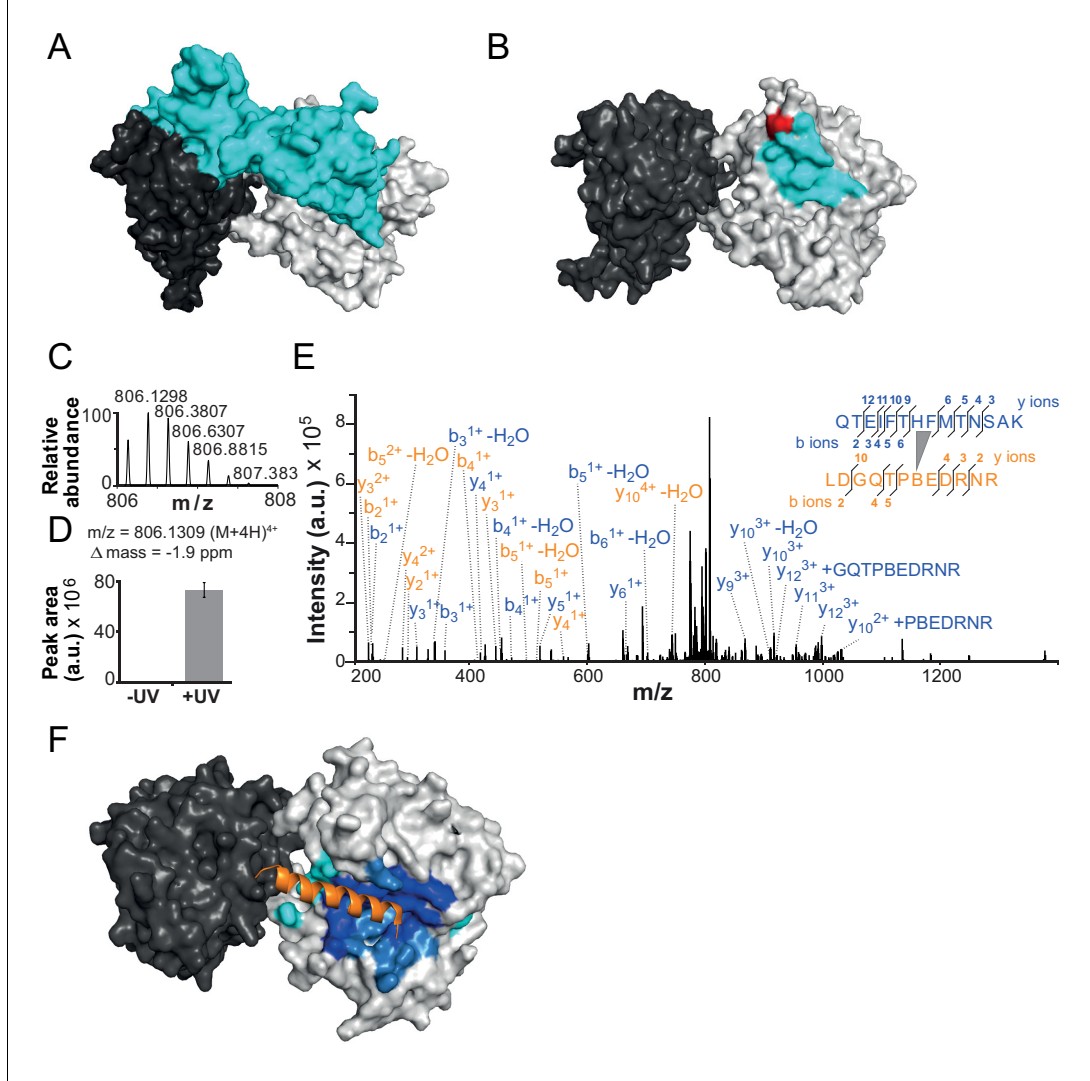

**Figure 4.** The NTR contacts Lobe 2 of the ATPase domain. (**A**) Surface representation of the Chd1 crystal structure (PDB code 3MWY) (*Hauk et al., 2010*). ATPase Lobe 1 and 2 are colored dark and light grey, respectively, and the N-terminal chromo domains cyan. (**B**) Homology model of the ISWI ATPase domain (*Forné et al., 2012*). Cyan: hypothetical binding interface of the ISWI NTR (see main text), red: position of Bpa substitution (H483). (**C**–**E**) Mass spectrometric validation of the crosslink XL1 (*Table 1*) formed between Bpa at position 483 and an NTR peptide. (**C**) Isotopic distribution of the crosslinked peptide. (**D**) UV-dependent increase of the signal for the crosslinked peptide. Extracted ion chromatograms of the ions were used for the quantification. (**E**) High resolution, high accuracy MS2 fragmentation spectrum. Top right: summary of observed product ions mapped onto the sequence of the crosslinked peptide. B: Bpa. (**F**) Predicted docking interface of AcidicN (blue and dark blue), AutoN (cyan and dark blue) and overlapping regions (dark blue) in the structural model of ISWI. The predicted interface for AcidicN overlaps with the interface for the acidic helix of the N-terminal chromo domains of Chd1 (orange) (*Hauk et al., 2010*).

The following figure supplements are available for figure 4:

**Figure supplement 1.** The effect of the H483B mutation on chromatin remodeling.

**Figure supplement 2.** Validation of additional crosslinks detected in the ISWI$_{H483B}$ dataset.

**Figure supplement 3.** Structural predictions of NTR elements.

into this hypothetical binding interface in ISWI (H483B; *Figure 4B*, red) using established strategies (*Forné et al., 2012*; *Chin et al., 2002*).

We first tested whether mutagenesis of H483 impacted catalysis. The H483B mutation diminished the DNA- and chromatin-stimulated ATPase activity of full-length ISWI by fourfold each, a result that may not be surprising given that the mutation is located in the conserved 'block D' of Snf2 ATPases (*Flaus et al., 2006*) (*Figure 6—figure supplement 4*). Importantly, the remodeling activity of ISWI$_{H483B}$ was reduced to a similar degree (threefold), indicating that the efficiency of remodeling per hydrolyzed ATP was unchanged (*Figure 4—figure supplement 1*). We conclude that ISWI$_{H483B}$, albeit hydrolyzing ATP more slowly than ISWI$_{WT}$, efficiently coupled ATP hydrolysis to chromatin remodeling, which suggested that the mutant remained structurally largely intact. Also, auto-regulation of ISWI$_{H483B}$ by its NTR was unperturbed because mutagenesis of the NTR had analogous effects on ISWI$_{WT}$ and ISWI$_{H483B}$ (see below), further justifying the use of ISWI$_{H483B}$ for crosslinking experiments.

Crosslinking of full-length ISWI$_{H483B}$ was induced by UV irradiation, and the crosslinks were mapped by high accuracy mass spectrometry (MS) (*Forné et al., 2012*; *Mueller-Planitz, 2015*). Remarkably, ISWI$_{H483B}$ crosslinked to several positions in the NTR within or adjacent to the ppHSA motif (*Figure 1A*, arrows; *Figure 4C–E*; *Figure 4—figure supplement 2*; *Table 1*). We independently replicated these crosslinking results with a truncated form of ISWI (ISWI$_{26-648}$), which lacked the HSS domain and non-conserved N-terminal amino acids (data not shown).

In our previous work, we incorporated Bpa in a variety of places on Lobes 1 and 2 of the ATPase domain but never observed crosslinks to the NTR (*Forné et al., 2012*; *Mueller-Planitz, 2015*). We therefore suggest that the ppHSA motif specifically docked to a location in proximity of amino acid 483 in Lobe 2. Docking of the NTR against Lobe 2 may be necessary for the structural integrity of ISWI-type remodelers (see above). The presence of ppHSA in other remodelers (Snf2, Lsh and Ino80; *Figure 1F*) predicts similar functions beyond the ISWI family.

If the NTR is structurally close to Lobe 2 of the ATPase module, AutoN and the neighboring AcidicN motif may also be able to contact Lobe 2. To explore this idea, we performed in silico docking studies to predict the binding site of AutoN and AcidicN. We carried out three independent docking runs to model the interaction of Lobe 2 with AutoN, AcidicN and AutoN-AcidicN, respectively (see Materials and methods for details). All three *ab initio* docking runs yielded a large cluster of models that identified the preferred binding site for AutoN and AcidicN (*Figure 4F*; *Figure 4—figure supplement 3A*). Docking of scrambled peptides as a control partially diminished the preference for this binding pocket (data not shown). Docking of AutoN-AcidicN against a homology model comprising both ATPase lobes gave very similar results, suggesting specificity of the motifs for binding to Lobe 2 (*Figure 4—figure supplement 3B*). We validated the docking results by mutagenesis further below.

Strikingly, AcidicN, which is predicted to be α-helical (*Figure 4—figure supplement 3C,D*), contacted Lobe 2 precisely where an acidic helix of the chromo domains of Chd1 bound (*Hauk et al., 2010*), which suggested conservation of this binding mode. Based on our results, we propose the

**Table 1.** Overview of crosslinks formed by ISWI$_{H483B}$.

| ID | Mass (D a) | Error (ppm) | Bpa peptide Sequence[*],[†] | Site | Target peptide Sequence[†] | Site |
|----|-----------|-------------|-------------|------|-----------------|------|
| XL1 | 3220.4946 | −1.9 | LDGQTPBEDRNR | 483 | QTEIFT<u>H</u>FM<sup>ox</sup>TNSAK[‡] | 59–60 |
| XL2 | 3204.5056 | −3.7 | LDGQTPBEDRNR | 483 | QTEIFTH<u>FM</u>TNSAK[‡] | 60–61 |
| XL3 | 2950.3474 | −1.0 | LDGQTPBEDR | 483 | QT<u>EI</u>FTHFM<sup>ox</sup>TNSAK[‡] | 55–59 |
| XL4 | 2934.3571 | −2.6 | LDGQTPBEDR | 483 | QTEIFT<u>H</u>F<u>M</u>TNSAK[‡] | 59–61 |
| XL5 | 2207.0968 | +0.2 | LDGQTPBEDRNR | 483 | SP<u>TK</u>PK[‡] | 69–72 |
| XL6 | 1936.9645 | −6.1 | LDGQTPBEDR | 483 | SPTK<u>P</u>K[‡] | 71–72 |
| XL7 | 1736.8594 | −6.1 | LDGQTPBEDR | 483 | GR<u>P</u>K | 75 |

[*]B symbolizes Bpa.

[†]Crosslinked amino acids are underlined; <sup>ox</sup>indicates oxidized methionine (+15.9949 Da).

[‡]Precise attachment sites not distinguishable from data.

NTR to adopt a structural architecture akin to the chromo domains of Chd1 (*Figure 4A*) despite complete lack of sequence conservation between the two.

## The H4 tail binds Lobe 2 adjacent to AutoN-AcidicN

Due to sequence similarity, the H4 tail and AutoN may compete for the same binding site (*Hwang et al., 2014*; *Clapier and Cairns, 2012*). We thus set out to identify the H4-tail binding pocket within ISWI and compare it to the predicted AutoN interaction surface.

We adopted two complementary crosslinking approaches. First, we used two different H4-tail peptides, which carried a Bpa moiety either at amino acid 1 or 10 (T1B and L10B peptides, respectively), and bound these peptides to ISWI$_{26-648}$ in the presence of DNA (*Mueller-Planitz et al., 2013*). After irradiation, a lower-mobility band was detected by SDS-PAGE, which suggested successful crosslinking (*Figure 5—figure supplement 1A,E*). We mapped several crosslinks of the H4 peptides to Lobe 2 by MS (*Figure 5—figure supplement 1A–F*; *Table 2*). Control experiments showed that the T1B H4 peptide stimulated the ATPase activity like a wild-type H4 peptide (*Figure 5—figure supplement 2A*).

Because the peptides may not exclusively bind ISWI in the physiological binding pocket, we pursued a second approach. We reconstituted entire nucleosomes bearing a photo-reactive benzophenone on the N-terminal tail of H4. Benzophenone labeling was achieved by chemical modification of single cysteine mutants of H4 (T1C and L10C). These nucleosomes bound to full-length ISWI and stimulated its ATPase activity like wild-type nucleosomes (*Figure 5—figure supplement 2B,C*) suggesting that they were properly recognized by the remodeler. UV-irradiation of full-length ISWI bound to benzophenone-labeled T1C nucleosomes retarded the mobility of the remodeler during SDS-PAGE, indicative of successful crosslinking (*Figure 5A*). MS analysis mapped a crosslink to Lobe 2 of ISWI (*Figure 5B–D*). We repeated these crosslinking experiments with the human ISWI homolog SNF2H. Both T1C- and L10C-labeled nucleosomes crosslinked to Lobe 2 of SNF2H (*Figure 5—figure supplement 1G–J*). In summary, two very different crosslinking approaches, one employing Bpa-containing peptides and one using benzophenone-derivatized nucleosomes, consistently yielded crosslinks between the H4 tail and Lobe 2 of the ATPase domain. *Table 2* lists all crosslink candidates, classified in terms of their reliability (see Materials and methods). Notably, methionine residues were overrepresented as targets of the photo-crosslinking approach, consistent with the known preference of benzophenones for methionine (*Wittelsberger et al., 2006*). In summary, our data strongly indicated the H4-tail binding site to reside on or close to Lobe 2.

To identify the H4-tail binding pocket we turned to crosslink-guided in silico docking of the H4-tail peptide. We only used the five crosslinks for this analysis that passed stringent quality controls (*Table 2*, high reliability; see also Materials and methods). The predicted docking interface is

**Table 2.** Overview of H4-tail mediated crosslinks.

| ID | Reliability | H4 | Remodeler construct | Mass (Da) | Error (ppm) | H4 peptide | | Remodeler peptide | |
| | | | | | | Sequence[*] | Site | Sequence[†] | Site |
|---|---|---|---|---|---|---|---|---|---|
| XL11 | high | nucleosomal | ISWI$_{WT}$ | 2034.8571 | −0.2 | XGR | 1 | QIQEFN<u>M</u>DNSAK | 495 |
| XL12 | high | nucleosomal | SNF2H | 2251.9753 | −0.4 | GXGK | 10 | VLDILEDYC<u>M</u>WR | 520 |
| XL13a | high | peptide | ISWI$_{26-648}$ | 1648.7601 | −0.4 | BGR | 1 | LDGQT<u>P</u>HEDR | 482 |
| XL13b | high | peptide | ISWI$_{26-648}$ | 1918.9052 | −0.9 | BGR | 1 | LDGQT<u>P</u>HEDRNR | 482 |
| XL13c | high | peptide | ISWI$_{26-648}$ | 3340.5374 | −2.8 | BGR | 1 | LDGQT<u>P</u>HEDRNRQIQEFNMDNSAK | 482 |
| XL14 | medium | nucleosomal | SNF2H | 2222.9624 | −1.5 | XGR | 1 | VLDILEDYC<u>M</u>WR[‡] | 519–22 |
| XL15 | medium | peptide | ISWI$_{26-648}$ | 1257.6261 | +2.3 | BGR | 1 | <u>M</u>VIQGGR | 578 |
| XL16 | medium | peptide | ISWI$_{26-648}$ | 1424.7832 | −3.9 | BGR | 1 | IVE<u>R</u>AEVK | 568 |
| XL17 | medium | peptide | ISWI$_{26-648}$ | 1453.7998 | −4.6 | GBGK | 10 | IVE<u>R</u>AEVK | 568 |

[*]B symbolizes Bpa; X symbolizes Benzophenone-labeled cysteine.

[†]Crosslinked amino acids are underlined.

[‡]Precise attachment sites not distinguishable from data.

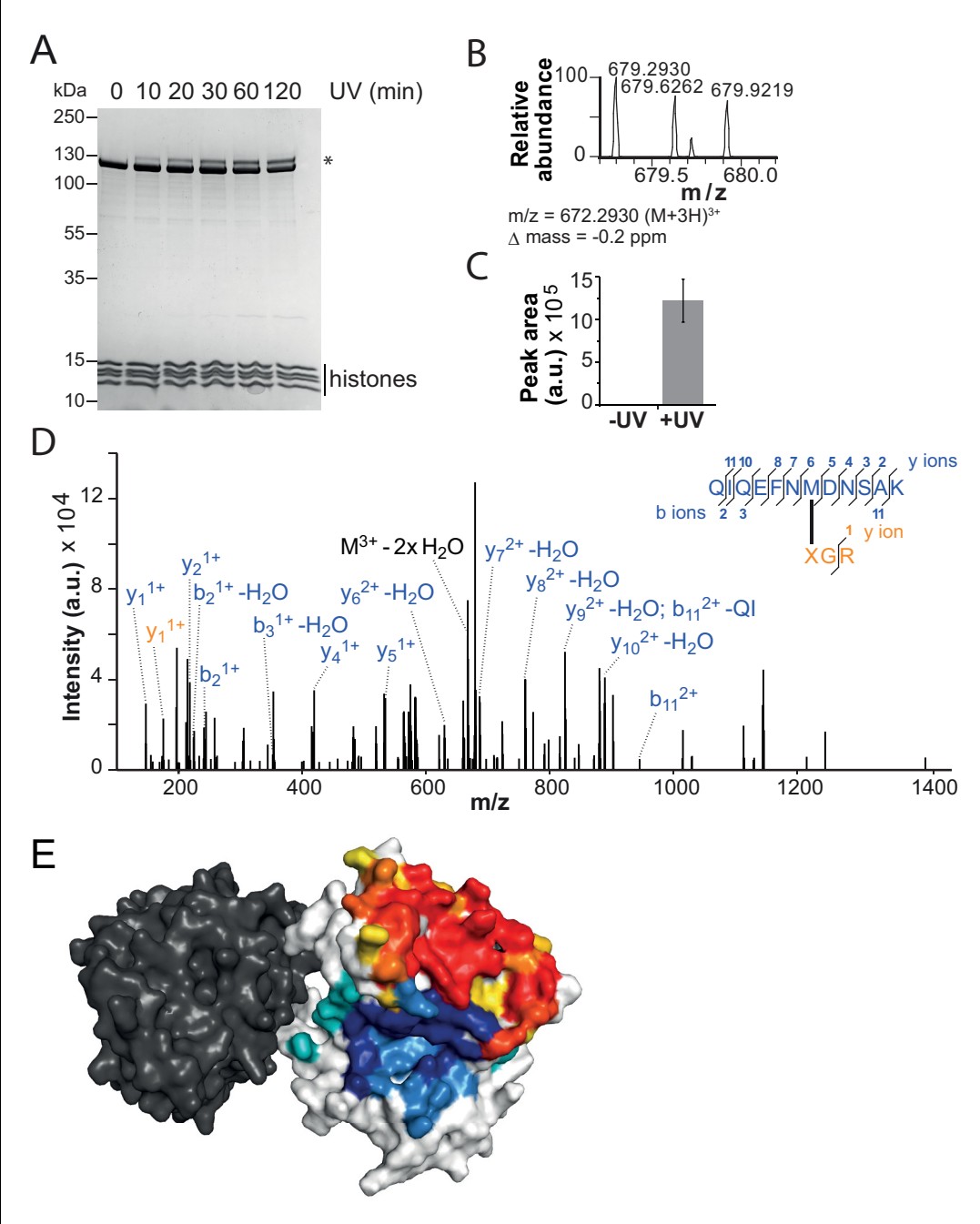

**Figure 5.** The binding sites of the NTR and the H4-tail on Lobe 2 are proximal. (**A–D**) Crosslinking of nucleosomes containing benzophenone-labeled H4 to ISWI. (**A**) Crosslinking time course analyzed by SDS-PAGE and Coomassie staining. The asterisk marks a UV-irradiation dependent band of lower mobility containing the crosslink mapped in **B–D**. (**B–D**) Mapping and validation of a crosslink (XL11; **Table 2**) formed in the upshifted band in **A**. Isotopic distribution of the crosslinked peptide, MS2 spectrum and quantification as in **Figure 4**. (**E**) Crosslink-guided in silico docking of an H4 peptide (amino acids 1–20) to ISWI. The predicted docking interface of the H4 tail on Lobe 2 is illustrated in a yellow and red color scale, which indicates low to high contact probabilities between the docked H4 tail and Lobe 2. The contact probabilities were calculated from a family of 383 docked structures (see Materials and methods). For comparison, the predicted docking interface of the NTR is shown in shades of blue (see **Figure 4F**).

The following figure supplements are available for figure 5:

*Figure 5 continued*

**Figure supplement 1.** Additional crosslinks between the H4 tail and ISWI or SNF2H.

**Figure supplement 2.** Controls for possible adversary effects of covalent modifications of the H4 tail.

**Figure supplement 3.** Surfaces on Lobe 2 that were sampled by selected amino acids in the H4 tail during crosslink-guided structural docking.

visualized in *Figure 5E* and *Figure 5—figure supplement 3*. Note that not all lower quality cross-links were compatible with this binding mode, possibly because the H4-tail peptide bound flexibly or in multiple binding modes (*Racki et al., 2014*). Some of these modes may not be strongly populated or functionally active as crosslinking can in principle trap fleeting intermediates. We also cannot rule out false positives among the lower quality candidates.

## AcidicN helps ISWI to recognize chromatin

Interestingly, the predicted docking interface of AcidicN was in close proximity to the H4-tail interface. This prompted us to investigate the function of AcidicN and – in the following section – its potential involvement in the H4-tail recognition process.

To study its function, we replaced three or six negatively charged amino acids in AcidicN by uncharged ones using conservative E to Q and D to N mutations. These mutants were denoted ISWI$_{+3}$ and ISWI$_{+6}$ respectively (*Figure 6A*). To improve solubility, ISWI$_{+6}$ was fused to a solubility tag (Z$_2$-tag; *Figure 6—figure supplement 1*). Control experiments ruled out interference of the Z$_2$-tag on catalytic properties of ISWI (*Figure 6—figure supplement 2A,B*).

Of note, the +3 and+6 mutants had a strongly deregulated ATPase, hydrolyzing ATP markedly faster than ISWI$_{WT}$ when presented with saturating amounts of naked DNA. In fact, DNA-stimulated ATPase rates of ISWI$_{+6}$ reached values of nucleosome-stimulated ISWI$_{WT}$ rates. Also its basal ATPase activity was strongly (20-fold) upregulated compared to ISWI$_{WT}$ (*Figure 6B*; *Figure 6—figure supplement 3A,B*). To rule out that co-purifying contaminating ATPases overwhelm the ATPase signal, we combined the +6 mutation with a point mutation in the ATPase that abrogates ATPase activity (E257Q). ATP hydrolysis and remodeling were negligible for ISWI$_{+6; E257Q}$, providing strong evidence against this possibility (*Figure 6—figure supplement 2B,C*).

In contrast to the DNA-stimulated reaction, nucleosome-stimulated ATPase and remodeling activities were comparable between the AcidicN mutants and ISWI$_{WT}$ (*Figure 6B,C*). Taken together, these results indicated that the AcidicN mutants were not simply hyperactive, but misregulated instead. More specifically, mutation of AcidicN prevented ISWI from properly recognizing whether chromatin was bound and led to futile ATP hydrolysis in the absence of chromatin.

To independently test this conclusion and to further validate the usefulness of the H483B mutant used further above, we combined the H483B and AcidicN mutations (*Figure 6—figure supplement 4*). DNA-stimulated ATP hydrolysis was strongly upregulated in the ISWI$_{+3; H483B}$ and ISWI$_{+6; H483B}$ double mutants relative to the ISWI$_{H483B}$ single mutant and reached levels of the chromatin-stimulated reaction. These data closely paralleled and therefore independently validated our results obtained with ISWI$_{+3}$ and ISWI$_{+6}$. We conclude that AcidicN regulates ISWI$_{WT}$ and ISWI$_{H483B}$ in a very similar fashion, further justifying the use of ISWI$_{H483B}$ for crosslinking experiments above.

To validate the predicted binding interface of AcidicN on Lobe 2 and to further probe the functionality of this interaction, we introduced mutations in Lobe 2. We selected three positively charged residues for mutagenesis, K403, R458 and R508, which are predicted to participate in docking to the negatively charged AcidicN motif (*Figure 7A*; *Figure 7—figure supplement 1*). Charge-reversal of these residues would be expected to weaken docking of AcidicN to Lobe2 and – in the simplest case – phenocopy the effects of the mutation of AcidicN. Indeed, the interface mutants had a strongly upregulated DNA-stimulated ATPase activity whereas chromatin-stimulated ATP turnover and nucleosome remodeling were largely unaffected (*Figure 7B,C*; *Figure 7—figure supplement 2*). The interface mutants therefore behaved just like the AcidicN mutants discussed above. A control mutant (ISWI$_{R486; 488D}$), carrying amino acid substitutions just outside of the predicted AcidicN binding interface, however, retained its ability to discriminate chromatin over DNA in the ATP hydrolysis

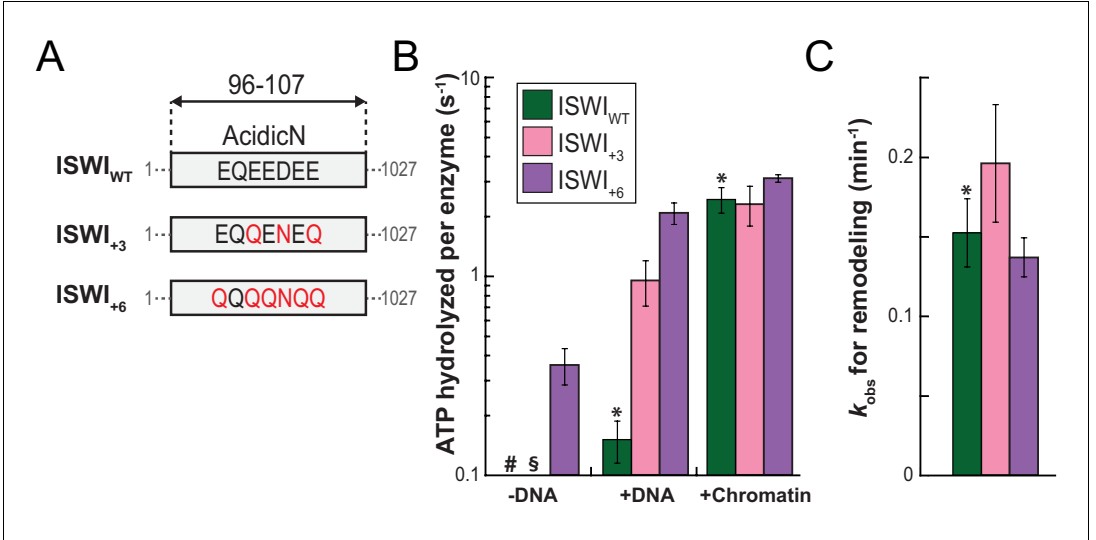

**Figure 6.** AcidicN is a strong negative regulator of the ATPase. (**A**) Design of AcidicN derivatives of ISWI (see also *Figure 6—figure supplement 1A*). (**B**) Effects of AcidicN mutation on ATP hydrolysis in absence or presence of saturating concentrations of DNA and chromatin. In absence of DNA, ATPase activities of $ISWI_{WT}$ (#) and $ISWI_{+3}$ (§) were $\leq 0.06$ $s^{-1}$. Errors are s.d. ($n \geq 4$). (**C**) Effects of AcidicN mutation on the remodeling activities. Nucleosomal arrays containing wild-type H4 were used. Errors are s.d. ($n \geq 3$) except for $ISWI_{+3}$ for which minimal and maximal values of two independent measurements are shown. Color code as in panel (**B**). Raw data of the remodeling assay can be found in *Figure 8—figure supplement 1*. Results for $ISWI_{WT}$ (*) are replotted for comparison from *Figure 3B,C*.

The following figure supplements are available for figure 6:

**Figure supplement 1.** AcidicN and AutoN mutants.

**Figure supplement 2.** Comparison of ATPase and remodeling activities of ISWI control variants used in this study.

**Figure supplement 3.** Saturation controls for $ISWI_{+6}$ in ATPase assays.

**Figure supplement 4.** AcidicN mutations upregulate the ATPase activity of $ISWI_{H483B}$.

assay (*Figure 7B*). These data support the notion that AcidicN interacts with Lobe 2 at the predicted interface and that this interaction is functionally important to discriminate whether chromatin is bound to the enzyme.

## AcidicN and AutoN cooperate during recognition of chromatin and H4 tail

To explore whether AcidicN takes part in H4-tail recognition, we measured the dependence of AcidicN mutants on the H4 tail in remodeling assays. Strikingly, the +3 and+6 ISWI derivatives lost most of their reliance on the H4 tail during remodeling (*Figure 8A*). In contrast, $ISWI_{\Delta ppHSA}$ and $ISWI_{\Delta ppHSA;\ \Delta AT-hook}$ retained a strong H4-tail dependence, which indicated that ppHSA had little involvement in H4-tail recognition.

Two AcidicN interface mutants described above (K403D and R458D) also depended less on the H4 tail during remodeling than $ISWI_{WT}$ (*Figure 8A*). The third mutant (R508D) and the control mutant (R485; 488D) were apparently still sensitive towards loss of the H4 tail. These two mutants, however, were not saturated with tail-less chromatin so that the calculated values represented upper limits for the H4-tail dependence (<24 fold and <15 fold, respectively; *Figure 8—figure supplement 1E* and data not shown).

Lack of the H4-tail dependence of AcidicN mutants was reminiscent of the phenotype previously described for the $ISWI_{2RA}$ mutation in AutoN (*Clapier and Cairns, 2012*). The 2RA mutation

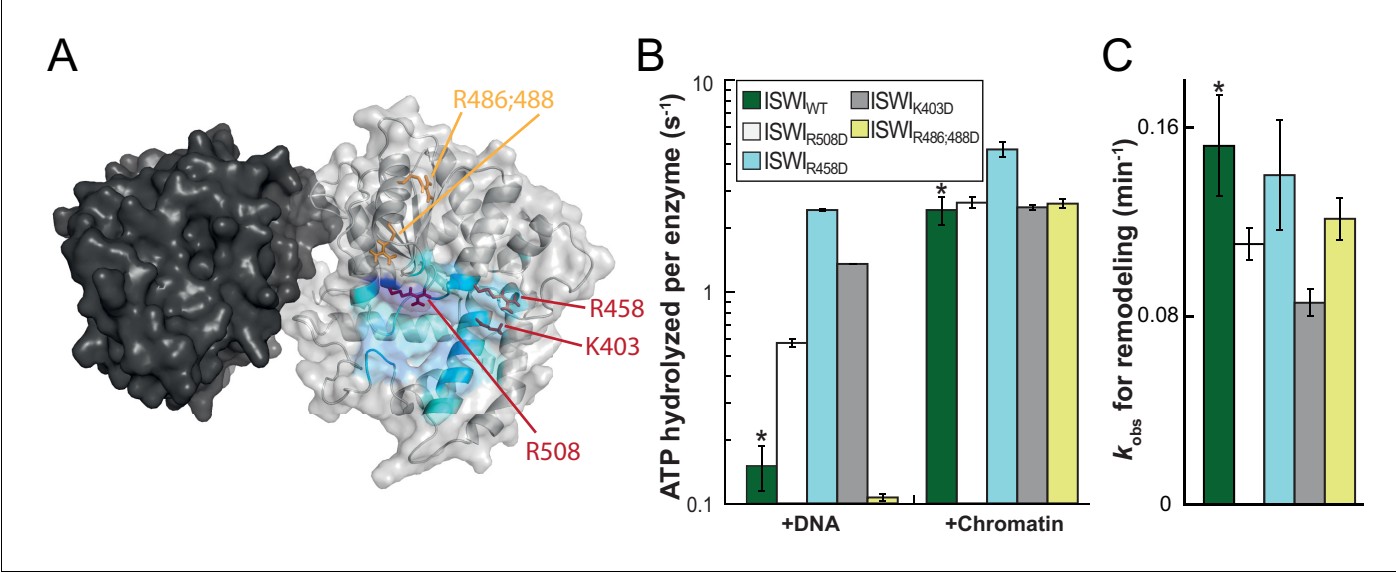

**Figure 7.** Validation of the predicted binding interface of AcidicN on Lobe 2. (**A**) Homology model of the ISWI ATPase domain. Dark and light grey, ATPase lobes 1 and 2, respectively; blue, hypothetical binding interface of AcidicN as in *Figure 4—figure supplement 3A*. Positively charged residues selected for mutagenesis are shown in red (AcidicN interface mutant) and orange (control mutant). (**B**) Mutation of the AcidicN interface (K403D, R458D and R508D) strongly upregulated DNA-stimulated ATP hydrolysis relative to ISWI$_{WT}$, whereas the nucleosome-stimulated ATP turnover was similar. In contrast, a control mutation (R486; 488D) had little effect on ATP hydrolysis. Saturating concentrations of DNA and chromatin were used. Errors are s.d. for ISWI$_{WT}$ and minimal and maximal values of two independent measurements for all other constructs. (**C**) AcidicN interface variants of ISWI robustly remodeled nucleosomes within twofold of ISWI$_{WT}$. Nucleosomal arrays containing wild-type H4 were used. Errors are s.d. (n ≥ 3) for ISWI$_{WT}$ and minimal and maximal values of two independent measurements for all other constructs. Raw data of the remodeling assay can be found in *Figure 7—figure supplement 2*. Color code as in (**B**). Results for ISWI$_{WT}$ (*) were replotted for comparison from *Figure 3B,C*.

The following figure supplements are available for figure 7:

**Figure supplement 1.** Coomassie-stained SDS-PAGE of purified recombinant ISWI constructs analyzed in *Figure 7*.

**Figure supplement 2.** Determination of rate constants for remodeling of AcidicN interface mutants.

(*Figure 6—figure supplement 1*) suppressed the dependence on the H4 tail also in our experiments, albeit our quantitative analysis showed an even more robust reduction than previously seen (*Figure 8A*). ISWI$_{2RA}$ was catalytically fully active, as was an AcidicN and AutoN double mutant (ISWI$_{+6; 2RA}$; *Figure 8—figure supplement 1*). Like the respective single mutants, ISWI$_{+6; 2RA}$ barely relied on the presence of the H4 tail (*Figure 8A*; *Figure 8—figure supplement 1*).

Compared to the respective single mutants, the ISWI$_{+6; 2RA}$ double mutant hydrolyzed ATP even faster in the absence of any ligand (*Figure 8B*). This result suggested that both motifs contributed to repression of the basal ATPase activity. In contrast, DNA- and chromatin-stimulated ATP turnover rates were not further perturbed by the double mutation (*Figure 8B*), consistent with both motifs cooperating during discrimination of chromatin from DNA.

## Discussion

Dozens of ATP-dependent chromatin remodeling factors are at work in any eukaryotic cell. Their activities impact every process that involves the cell's genetic material, including transcription, replication, DNA repair and recombination. Dysfunction and improper regulation of these complexes may have dire consequences for human health (*Kadoch and Crabtree, 2015*; *Garraway and Lander, 2013*). Perhaps as a consequence, remodelers across many families independently evolved intricate mechanisms for autoregulation (*Clapier and Cairns, 2012*; *Hauk et al., 2010*; *Wang et al., 2014*; *Clapier et al., 2016*; *Gottschalk et al., 2009*).

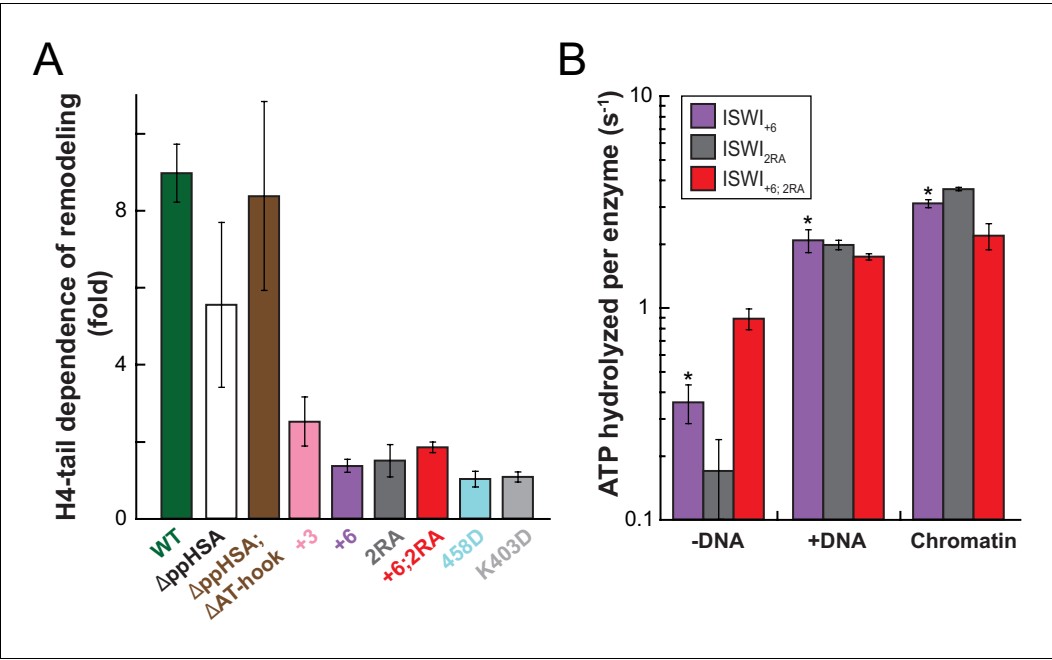

**Figure 8.** Mutation of AcidicN, the AcidicN binding interface or AutoN suppresses dependence on the H4-tail. (**A**) H4-tail dependence of the remodeling activities of ISWI variants. Values were calculated from the observed remodeling rate constants obtained for WT and tail-less H4 chromatin (*Figure 8—figure supplement 1E*). (**B**) ATP hydrolysis measurements of ISWT$_{+6}$, ISWI$_{2RA}$ and ISWI$_{+6;\ 2RA}$ in absence or presence of saturating concentrations of DNA and chromatin.

The following figure supplement is available for figure 8:

**Figure supplement 1.** Raw data of the remodeling assays.

It has been known for many years that the activity of ISWI remodelers is regulated by the H4 tail (*Clapier et al., 2001*). Regulation by the H4 tail was later also discovered for remodelers of the Chd1 (*Ferreira et al., 2007*) and Alc1 families (*Ahel et al., 2009*). The molecular mechanism of H4-tail recognition and regulation has remained elusive, not least because the tail's binding site had not been mapped. Using crosslinking-MS, we found the H4 tail to bind to the conserved Lobe 2 of the ATPase module. Direct binding to the ATPase domain explains regulation of otherwise divergent remodeler families and explains the influence of the H4 tail on catalytic, as opposed to purely binding steps (*Clapier et al., 2001*; *Dang et al., 2006*). Our data do not rule out additional binding sites on other domains and on ISWI's partner subunit ACF1 as proposed earlier (*Boyer et al., 2004*; *Grüne et al., 2003*; *Hwang et al., 2014*).

ISWI and Chd1 proteins have evolved a complex autoregulatory mechanism. This mechanism involves an autoinhibitory domain N-terminal to the ATPase. Inhibition by this domain is countered in an unknown fashion by H4-tail binding. Two limiting scenarios can explain the data (*Figure 9*).

The first model has been proposed earlier (*Clapier and Cairns, 2012*) and posits that AutoN acts as a pseudosubstrate by mimicking part of the basic patch of the H4 tail. In fact, AutoN (amino acids 'RHRK', which are present in many but not all ISWI proteins; *Figure 1D* and *Figure 1—figure supplement 1*) was initially discovered by way of its resemblance to the amino acids 'R$_{17}$H$_{18}$R$_{19}$K$_{20}$' on histone H4 (*Clapier and Cairns, 2012*). In this model, the basic patch of the H4 tail must compete with AutoN for the same binding site on the ATPase domain, such that AutoN and possibly the entire NTR is displaced upon tail binding (*Clapier and Cairns, 2012*; *Hauk et al., 2010*). This model is supported by the observations that the NTR can in principle undergo conformational changes (*Mueller-Planitz et al., 2013*) and that the chromo domains of Chd1 must rearrange before the ATPase domain assumes a catalytically active conformation (*Hauk et al., 2010*). Direct experimental support for a shared binding site of AutoN and H4 basic patch has been lacking, however, and the

resemblance of the two motifs may be purely coincidental in principle. Of the four amino acids that resemble the H4 tail, only three ($R_{17}H_{18}R_{19}$) were found to be functionally important for ISWI enzymes (*Fazzio et al., 2005*; *Clapier et al., 2002*; *Clapier and Cairns, 2012*). Recent crystallographic evidence also did not support the molecular mimicry hypothesis (see below) (*Yan et al., 2016*).

We favor a second, simpler model, which does not invoke molecular mimicry (*Figure 9*). In this model, the AutoN and the H4-tail binding sites are not identical, possibly allowing simultaneous binding of both to Lobe 2 at least temporarily. This scenario is fully compatible with our suggestion that the docking sites for the H4 tail and AutoN-AcidicN are adjacent to each other but not overlapping (*Figure 5E*). Conceivably, the negatively charged AcidicN motif may even promote binding of the basic H4 tail to a neighboring site. A structural rearrangement of the NTR upon H4 tail binding is compatible with but not required in this model. Similarly, conformational changes of the NTR upon DNA binding (*Mueller-Planitz et al., 2013*; *Hauk et al., 2010*) or during other steps of the reaction cycle are also fully consistent with it.

Intriguing parallels between ISWI's NTR and Chd1's chromo domains become apparent. Our crosslinking results indicate that the NTR of ISWI docks against Lobe 2 of the ATPase domain in a very similar fashion as the chromo domains of Chd1, and docking appears to involve an acidic motif in both cases (*Hauk et al., 2010*). Thus, the overall conformational architecture of ISWI's ATPase module may be shared with Chd1. Moreover, both domains are known to inhibit the ATPase, both are predicted to undergo conformational changes upon substrate binding (*Mueller-Planitz et al., 2013*; *Clapier and Cairns, 2012*) and both confer sensitivity towards the histone H4 tail (*Clapier and*

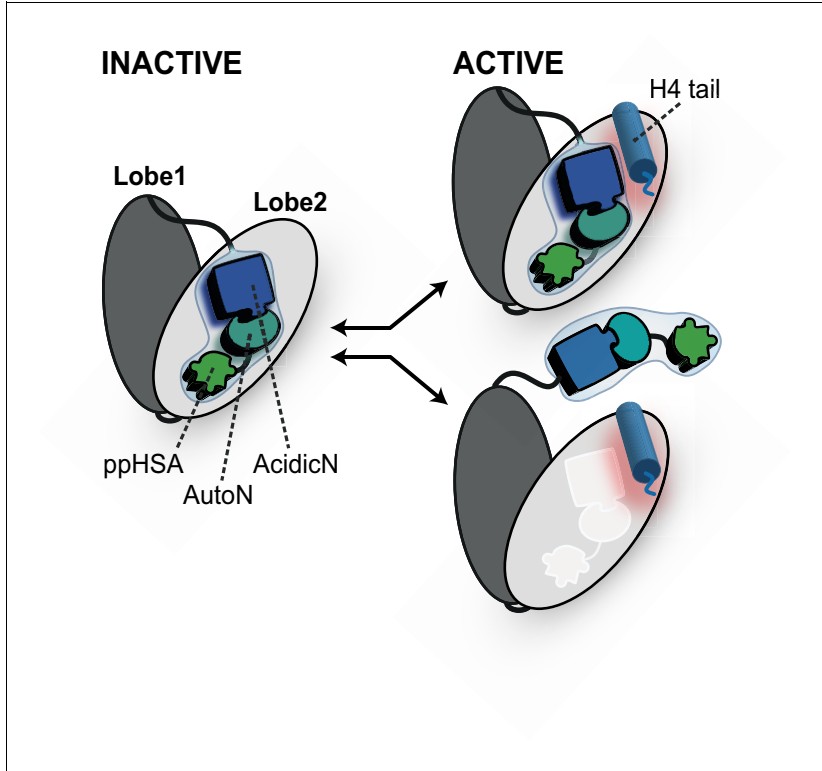

**Figure 9.** Proposed models for autoregulation imposed by the NTR and the recognition process of the H4 tail. The ppHSA motif, AcidicN and AutoN dock against Lobe 2 of the ATPase domain, promoting an overall structural architecture of the ATPase module that is reminiscent of Chd1 (*Figure 4A*). AcidicN and AutoN functionally collaborate in the H4 tail recogniton process. The docking site of AutoN-AcidicN is adjacent to the H4 tail potentially allowing simultaneous binding (top). Alternatively, the H4 tail may displace the NTR as suggested previously (bottom) (*Clapier and Cairns, 2012*).

*Cairns, 2012*; *Hauk et al., 2010*). Thus, despite complete lack of sequence conservation between both domains, they appear to have evolved very similar functionalities.

The NTR of ISWI contains several conserved motifs whose functions have mostly remained unexplored so far. Because the ppHSA motif and adjacent regions crosslinked to the ATPase lobe 2, we suggest that it is important for docking the NTR against the ATPase domain. Consistent with such a structural role of this motif, we found that ISWI$_{\Delta ppHSA}$ is destabilized in vitro and in vivo. Of note, the ppHSA motif is present in a wide variety of unrelated remodelers, including Ino80, Lsh, and Snf2, suggesting that their ATPases, too, might bind the ppHSA motif and assemble into a structurally analogous architecture.

In this study, we functionally characterized AcidicN, a novel motif in the NTR. ISWI with a mutated AcidicN had a deregulated, hyperactive ATPase activity. Notably, this mutant hydrolyzed ATP with comparable velocities when bound to either DNA or nucleosomes, indicating that it lost its ability to discriminate between them. In particular, it lost its H4-tail dependence. This phenotype is reminiscent of mutations in the acidic helix in Chd1 (*Hauk et al., 2010*), underscoring the functional parallels between the NTR and chromo domains discussed above. The effects of AcidicN mutations were also remarkably similar to AutoN mutations (*Clapier and Cairns, 2012*), which suggested that they work together. The mechanism of autoinhibition by the NTR therefore may involve more than simple mimicry of H4's basic patch by AutoN (*Clapier and Cairns, 2012*). Supporting its functional importance, AcidicN is at least as conserved as AutoN in our alignments.

During the revision of this manuscript, a crystal structure of the ATPase module of ISWI from a thermophilic fungus became available (*Yan et al., 2016*). Even though both studies relied on different approaches, they arrived at very similar conclusions. As suggested by our crosslinking and modeling data, the NTR packed against the ATPase domain in the structure of the thermostable ISWI. We correctly predicted the AcidicN binding pocket on Lobe 2 (*Figure 4—figure supplement 3A*), and our crosslinks between Lobe 2 and the NTR were fully supported by the structure as well. Finally, the authors succeeded in co-crystallizing a histone H4-tail peptide with Lobe 2 of the ATPase. Even though only the basic patch of the tail peptide was visible in the structure, its location overlapped well with the position of the modeled H4 basic patch (*Figure 5—figure supplement 3C*). AutoN crystallized in closer proximity to the interaction site of the basic patch than suggested by modeling, but molecular mimicry of AutoN with the basic patch was not supported by the structure.

Sensitive biophysical assays will be instrumental in the future for resolving conformational changes that may occur during H4-tail recognition and for understanding their functional importance in ISWI complexes. Moreover, ascertaining the predicted role of H4-tail recognition for the formation or maintenance of compact heterochromatic regions remains an important goal.

## Materials and methods

### Amino acid sequence alignments and sequence logos

Search for homologous proteins of full-length *Drosophila* ISWI and alignment of sequences were done using HHblits with standard settings. Sequence logos of conserved NTR motifs were derived with WebLogo three from this alignment (*Schneider and Stephens, 1990*). Proteins containing the ppHSA motif were identified by PSI-BLAST against the 120 N-terminal amino acids of ISWI. The alignment was done using T-Coffee. Sequence alignments were visualized using Jalview 2.9.

### Spotting assays

*S. cerevisiae* Isw1 alleles were cloned into selected destination vectors of a galactose-inducible hybrid promoter library (generously provided by Dr. Hal Alper, UT Austin, USA) (*Blazeck et al., 2012*). The following destination vectors were used, sorted according to increasing promoter strength: Gal4pBS2-P$_{leum}$ (denoted '+' in *Figure 2B*; *Figure 2—figure supplement 1*), Gal4pBS4-P$_{leum}$ ('++'), UAS$_{gal}$-A9-P$_{cyc}$ ('+++'), and UAS$_{gal}$-P$_{gal}$ ('++++'). Destination vectors were XbaI and ClaI digested and gel purified. Isw1 derivatives were PCR-amplified from yeast genomic DNA, gel purified and ligated into the destination vectors by Gibson assembly. All spotting assays employed untagged Isw1 variants. All constructs were sequence verified before transformation. As an empty vector control, the UAS promoter, coding and terminator sequences were removed from the Gal4pBS2-P$_{leum}$ plasmid by AscI and MluI digest and subsequent self-ligation.

YTT227 (TKO), YTT225 (DKO) and W1588-4c (wild-type; *Table 3*) were transformed with indicated plasmids via a standard transformation protocol. Single colonies were picked and grown overnight in Synthetic complete (SC)-Ura + Glucose (2%) media. The culture was then diluted to OD 0.05 in SC-Ura Galactose media (2%) and grown for 24 hr. Cells were diluted again to OD 0.1 in Galactose media and grown for another 24 hr before spotting. Cells were diluted to OD 1.0, and tenfold serial dilutions were spotted on galactose media and incubated at 30°C, 37°C or 38.5°C for 72 hr. At least two replicates were performed on different days with a single transformant of a sequence-verified clone.

## Western analysis

For Western analysis, Isw1 variants were C-terminally tagged by fusion to a cassette containing a $(GGS)_2$ linker, a 3C cleavage site, a $(GGS)_5$ linker and a TAP tag. YTT227 that expressed TAP-tagged Isw1 variants was induced with galactose as above, diluted to OD 0.1 and grown to OD 1.0 in 10 ml SC-Ura + Galactose media. YFMP047 (*Table 3*), containing a genomically TAP-tagged Isw1 allele, was grown as a control in YPAD media. Cells were harvested, washed twice with cold water and dissolved in 200 μl Extraction buffer (40 mM Hepes-KOH pH 7.5, 10% Glycerol, 350 mM NaCl, 0.1% Tween-20, 1 μg/ml Pepstatin, 2 μg/ml Leupeptin, 2 μg/ml Aprotinin, 1 mM PMSF. Glass beads (200 μl) were added, and the suspension was vortexed for 10 min with a 30 s on/off cycle on ice. After centrifugation (13,000 rpm, 20 min 4°C), supernatants were harvested, aliquoted (50 μl), flash frozen, and stored at −80°C for subsequent use. Supernatants were thawed on ice and 50 μg of each extract was loaded on a 10% SDS gel. Anti-TAP antibody (CAB1001, ThermoFisher; 1:5000 dilution) was used to detect TAP-tagged ISW1 mutants and anti-H3 antibody (ab1791, Abcam; 1:20,000 dilution) was used as a loading control. Membranes were scanned using the LI-COR Odyssey IR imaging system (ODY-0853) and bands were quantified using Image Studio Lite v5.2.5. Expression levels were normalized to the signal of genomically integrated TAP-tagged Isw1. Two technical replicates were performed.

## Construct design and cloning of *Drosophila* ISWI variants

A pPROEX-HTb–based expression plasmid with the gene encoding *Drosophila* $ISWI_{WT}$ (kindly provided by C. Müller; EMBL, Heidelberg, Germany) served as the template for all ISWI variants. An overview over cloned ISWI variants is presented in *Figure 3—figure supplement 1A* and *Figure 6—figure supplement 1A*. All ISWI genes were fused N-terminally to a $His_6$-tag. To generate $ISWI_{\Delta ppHSA}$ and $ISWI_{\Delta ppHSA; \Delta AT-hook}$, a 3C cleavage site was introduced at the desired site by Quik-Change mutagenesis or polymerase incomplete primer extension. The trigger factor gene was amplified from pTf16 (Takara Bio Inc.) and fused to the $ISWI_{WT}$ gene by Gibson assembly. $ISWI_{+6}$ was subcloned into the pET-Z2 plasmid (kindly provided by Dr. Arie Geerlof, Helmholtz Zentrum, Munich, Germany).

## Protein expression and purification of ISWI variants

Expression and purification of $His_6$-tagged $ISWI_{WT}$ and its derivatives was performed essentially as described (*Forné et al., 2012*) with the following variations. Tags or parts of the NTR were cleaved off by specific proteases (TEV and 3C, respectively) as indicated (*Figure 3—figure supplement 1A*; *Figure 6—figure supplement 1A*). $ISWI_{H483B}$ was expressed and purified as described (*Forné et al., 2012*). During its purification, the UV light of the FPLC remained switched off to protect the Bpa residue. All ISWI variants were purified once except $ISWI_{+3}$ and $Z_2-ISWI_{+6}$, which were purified twice.

**Table 3.** Yeast strains used in this study.

| Strain | Genotype | Reference |
|---|---|---|
| W1588-4C | *MATa ade2-1 his3-11,15 leu2-3,112 trp1-1 ura3-1 can1-100 but RAD5* | *Tsukiyama et al. (1999)* |
| YTT227 | *MATa ade2-1 his3-11,15 leu2-3,112 trp1-1 ura3-1 can1-100 but RAD5 isw1::ADE2 isw2::LEU2 chd1::TRP1* | *Tsukiyama et al. (1999)* |
| YTT225 | *MATa ade2-1 his3-11,15 leu2-3,112 trp1-1 ura3-1 can1-100 but RAD5 isw2::LEU2 chd1::TRP1* | *Tsukiyama et al. (1999)* |
| YFMP047 | *MATa his3Δ1 leu2Δ0 met15Δ0 ura3Δ0 ISW1-TAP::HIS3MX6* | *Open Biosystems* |

The independent preparations were indistinguishable in ATPase assays ($ISWI_{+3}$ and $Z_2$-$ISWI_{+6}$). Whereas $ISWI_{+3}$ preparations were not directly compared, independent $Z_2$-$ISWI_{+6}$ preparations also yielded same results in remodeling assays.

## Expression and purification of SNF2H

A pBH4-based expression plasmid encoding full-length human SNF2H (kindly provided by G. Narlikar; UCSF, San Francisco) was transformed into Rosetta competent *E. coli* cells. Protein expression was performed in 2x YT medium (20 g/l tryptone, 10 g/l yeast extract, 10 g/l NaCl) supplemented with 34 mg/l chloramphenicol and 100 mg/l ampicillin. Expression of SNF2H was induced by addition of 0.4 mM IPTG at 18°C for approximately 18 hr. Bacteria cells were resuspended in 20 ml lysis buffer per 1 l culture (25 mM HEPES pH 8.0, 300 mM KCl, 7.5 mM imidazole, 10% glycerol, 1 mM DTT) supplemented with protease inhibitors (1 mM PMSF, 1 mg/l Aprotinin, 1 mg/l Leupeptin, 0.7 mg/l Pepstatin) per 1 l culture, and lysed by French Press (Thermo Spectronic) and ultrasonication (Branson). Per 1 l lysed bacteria culture, 1000 U Benzonase (Merck Millipore) were added. The lysate was clarified by centrifugation (30 min, SS34 rotor). The N-terminal $His_6$-tagged SNF2H was purified by nickel affinity chromatography (HisTrap HP, 5 ml; GE Healthcare). An elution gradient was applied with 25 mM HEPES pH 7.0, 300 mM KCl and 400 mM Imidazole and enzyme-containing fractions were pooled. Contaminating DNA was removed by passing the sample over an anion exchange column (Mono Q 5/50 GL ion exchange column; GE Healthcare) that was pre-equilibrated in SEC buffer (25 mM HEPES pH 7.5, 300 mM KCl, 1 mM DTT). The flow-through of the column was collected. The protein sample was concentrated to 0.5–1 ml per 1 l of original *E. coli* culture in centrifugal filters (Amicon Ultra-4, 30 kDa MWCO; Millipore). TEV protease (prepared in-house) was added to a final concentration of 0.075–0.15 mg/ml and the concentrated protein sample was dialyzed against 1 l SEC buffer overnight in dialysis tubing (6000–8000 Da MWCO; Sectra/Por). The protein sample was loaded onto a size exclusion chromatography column (Superdex 200 HiLoad 16/60, 120 ml; GE Healthcare) pre-equilibrated in SEC buffer. Elution fractions were pooled according to purity and, as necessary, concentrated and dialyzed into storage buffer (25 mM HEPES pH 7.5, 210 mM KCl, 15% glycerol, 1 mM DTT) for at least 16 hr.

## Nucleosome reconstitution

*Drosophila* histones were purified as described (*Klinker et al., 2014*; *Luger et al., 1999*). The 187 bp long Widom-601 derivative used for end-positioned mononucleosomes (0N40) was excised from pFMP151 with SmaI (NEB) and PAGE purified. DNA for 25-mer nucleosomal arrays used in remodeling assays was excised from pFMP233 with EcoRI HF, HincII and AseI (NEB) and purified by phenol/chloroform extraction and ethanol precipitation. Polynucleosomes used in ATP-hydrolysis assays were assembled on linearized plasmid DNA (pT7 blue derivative). Histone octamers, mononucleosomes and polynucleosomes, including 25-mer nucleosomal arrays, were prepared by salt-gradient dialysis as described (*Mueller-Planitz et al., 2013*; *Luger et al., 1999*). Mononucleosomes were further purified by glycerol gradient ultracentrigation. Nucleosomal arrays were purified further by $Mg^{2+}$ precipitation (3.5 mM for WT-H4 arrays, 8.5 mM for H4-tail deleted arrays) (*Mueller-Planitz et al., 2013*). The concentration of nucleosomal DNA was determined by measuring its UV absorption at 260 nm. For nucleosomal arrays, concentrations refer to the concentration of individual nucleosomes.

## Enzyme assays

Remodeling and ATPase assays were performed in 25 mM HEPES-KOH, pH 7.6, 50 mM NaCl, 1 mM $MgCl_2$, 0.1 mM EDTA, 10% glycerol, 0.2 g/l BSA and 1 mM DTT at 26°C in the presence of a ATP regenerating system as described (*Mueller-Planitz et al., 2013*).

ATP hydrolysis was monitored by an NADH-coupled ATP hydrolysis assay (*Mueller-Planitz et al., 2013*; *Forné et al., 2012*). Saturating concentrations of ATP-$Mg^{2+}$ (1 mM) and of nucleic acids ligands were used (0.2 mg/ml of linearized pT7blue and 0.1 mg/ml of chromatin assembled on the same DNA, respectively). Saturation of DNA and chromatin was controlled by varying the concentration of the ligands at least 16-fold (*Figure 3—figure supplement 2*; *Figure 6—figure supplement 3*). Occasional occurrence of air bubbles in ATPase experiments precluded accurate measurements; affected samples were excluded from the analysis. In no other assays were outliers excluded.

Remodeling activity was probed by a restriction enzyme accessibility assay (*Mueller-Planitz et al., 2013*). A 25-mer nucleosomal array with a 197 bp nucleosomal repeat length was used. The 19th nucleosome of this array occluded a unique KpnI site at position $-32$ relative to its dyad (*Mueller-Planitz et al., 2013*). Arrays (100 nM) were incubated with ISWI derivatives at the indicated concentrations, ATP-Mg$^{2+}$ (1 mM) and KpnI (2 U/ml). Reactions were quenched with SDS (0.4%) and EDTA (20 mM) before the samples were deproteinized, ethanol precipitated and resolved by agarose gel electrophoresis (*Mueller-Planitz et al., 2013*). $k_{obs}$ for remodeling was obtained by fitting time courses to a single exponential function. When the enzyme concentration was varied $\geq$threefold, typically between 100 nM and 300 nM, similar values for $k_{obs}$ were obtained with a few exceptions, suggesting that arrays were generally saturated (*Figure 3—figure supplement 3*; *Figure 4—figure supplement 1*; *Figure 7—figure supplement 2*; *Figure 8—figure supplement 1A–D* and data not shown). The exceptions comprised ISWI$_{\Delta ppHSA;\ \Delta AT\text{-}hook}$, ISWI$_{+3}$ and ISWI$_{H483B}$ on WT-arrays and ISWI$_{+3}$, ISWI$_{R508D}$ and ISWI$_{R486;\ 488D}$ on tail-less H4 arrays.

## UV crosslinking

To site-specifically attach a UV-reactive benzophenone residue to full-length histone H4, single cysteines were introduced into the histone H4 tail by site directed mutagenesis at the indicated positions. 4-(N-Maleimido)benzophenone (Sigma) was dissolved to 100 mM in N,N-Dimethylformamide (DMF) and added to a final concentration of 3 mM to denatured single cysteine variants of H4 (1 mg/ml) in 20 mM Tris/HCl pH 7.1, 7 M Guanidine-HCl, 5 mM EDTA, 2 mM TCEP for 2 hr at room temperature. After a 3 hr incubation in the dark, the labeling reaction was stopped by adding 20 mM DTT for 20 min.

UV-Crosslinking was performed in uncoated 384-well plates or 96-well plates (Greiner) on ice using the 365 nm irradiation of a BioLink UV-Crosslinker (Peqlab) for the indicated durations. Crosslinking between benzophenone-labeled nucleosomes (0N40; 1 µM) and stoichiometric amounts of ISWI or SNF2H was performed in 20 mM Tris/HCl, pH 7.7, 100 mM KCl, 0.1 mM EDTA, 3 mM DTT. Crosslinking between ISWI$_{26\text{-}648}$ (0.1 mg/ml) and a histone H4 peptide comprising the 24 N-terminal amino acids of H4 carrying a Bpa substitution at position 1 or 10 was carried out in the presence of 13 µM 59 bp DNA duplex in 25 mM HEPES-KOH, pH 7.6, 50 mM NaCl, 1 mM MgCl2, 0.1 mM EDTA, 10% glycerol and 1 mM DTT for 3 hr as above. Samples were subsequently digested with benzonase before further processing. Crosslinking between Bpa variants of ISWI (H483B) was carried out as described (*Forné et al., 2012*).

UV-irradiated samples and unirradiated control samples were separated by SDS-PAGE and Coomassie stained. Protein bands were excised and trypsin digested for subsequent mass spectrometry as described (*Forné et al., 2012*; *Wilm et al., 1996*).

## Mapping of crosslinks by LC-MS/MS

For LC-MS/MS, 5 µl were injected in either an Ultimate 3000 system (Thermo) and desalted on-line in a C18 micro column (75 µm i.d. x 15 cm, packed with C18 PepMap, 3 µm, 100 Å by LC Packings) or desalted offline using C18 Stagetip and injected in an Ultimate 3000 RSLCnano system (Thermo). Desalted sample was then separated in a 15 cm analytical C18 micro column (75 µm i.d. packed with C18 PepMap, 3 µm, 100 Å by LC Packings or homepacked 75 µm ID with ReproSil-Pur C18-AQ 2.4 µm from Dr. Maisch) with a 40 to 60 min gradient from 5% to 60% acetonitrile in 0.1% formic acid. The effluent from the HPLC was directly electrosprayed into an LTQ-Orbitrap XL as described before (*Forné et al., 2012*) or a Q Exactive HF MS (Thermo). The Q Exactive HF MS was operated in a data-dependent mode. Survey full scan MS spectra (from m/z 375–1600) were acquired with resolution R = 60,000 at m/z 400 (AGC target of $3 \times 10^6$). The ten most intense peptide ions with charge states between 3 and 5 were sequentially isolated to a target value of $1 \times 10^5$, and fragmented at 27% normalized collision energy. Typical mass spectrometric conditions were: spray voltage, 1.5 kV; no sheath and auxiliary gas flow; heated capillary temperature, 250°C; ion selection threshold, 33.000 counts.

Each Thermo binary raw file was converted to a dta file using Decon2LS (*Zimmer et al., 2006*) or to an mgf file using Proteome Discoverer 1.4 (Thermo) and -as needed- recalibrated with the Post-Search Recalibrator Node. Crosslinks were mapped by Crossfinder (*Forné et al., 2012*; *Mueller-*

*Planitz, 2015*). Typical error windows were ±10 ppm for MS1 searches and ±15 ppm for MS2 searches. All amino acid residues were regarded as potential sites of crosslinking.

Crosslink candidates were independently validated by the authors J.L., S.P., N.H. and F.M.-P. and rated as high, medium and low confidence. The validation comprised a general assessment of the spectrum quality, removal of wrong product ion assignments, and evaluation of the actual evidence for the presence of the two peptides within the crosslink. The mass spectrometry data have been deposited to the ProteomeXchange Consortium via the PRIDE partner repository with the dataset identifier PXD005831.

## In silico docking studies

Interactions between NTR motifs and the histone H4 tail with Lobe 2 were modeled using the fully blind peptide-protein docking protocol pepATTRACT (*Schindler et al., 2015a*) in the ATTRACT docking engine (*de Vries et al., 2015*) (www.attract.ph.tum.de/peptide.html). The termini of the motifs ('peptides') were left uncharged, other parameters were set to the default values as described (*Schindler et al., 2015a*). Briefly, for each motif three idealized peptide conformations (extended, α-helical and poly-proline) were generated from sequence and this peptide ensemble was docked rigidly against the protein domain using the ATTRACT coarse-grained force field (*Zacharias, 2003*). The top-ranked 1000 structures were subjected to two stages of atomistic refinement using the flexible interface refinement method iATTRACT (*Schindler et al., 2015b*) and a short molecular dynamics simulation in implicit solvent with the AMBER program (*Case et al., 2014*). The pepATTRACT protocol requires neither knowledge about the peptide binding site nor of the bound peptide conformation and is therefore suitable for predicting complexes between proteins and motifs from intrinsically unstructured regions.

The structure of ISWI ATPase Lobe 2 (residues 352–637) was modeled by homology from the structure of Chd1 (PDB 3MWY) using MODELLER (*Webb and Sali, 2016*). We performed three docking runs modeling the potential binding site for the AutoN motif (residues 89–97; DHRHRKTEQ), the AcidicN motif (residues 96–104; EQEEDEELL) and the full module AutoN+AcidicN (residues 89–104; DHRHRKTEQEEDEELL) separately. During the first two runs, we modeled the peptide ensemble from the sequence as described above. For the third run, we used PEP-FOLD2 (*Shen et al., 2014*), PEP-FOLD3 (*Lamiable et al., 2016*) and I-TASSER (*Yang et al., 2015*) servers to predict the structure of the module. We used the resulting 13 conformations for ensemble docking to Lobe 2. To test the specificity of the docking solutions, we also modeled Lobe 2 binding to scrambled sequences of AutoN (HRQHKDERT), AcidicN (LEDELQEEE) and AutoN-AcidicN (HLREQLDTHE REDEKE). Docking of AutoN-AcidicN against the homology model comprising both ATPase lobes was done as described above.

During docking of the histone H4 tail (residues 1–20) to Lobe 2, we used the five high confidence crosslinks (*Table 3*), which – due to redundancy – provided three unique amino acid linkages. These linkages were used as upper harmonic distance restraints with a maximum distance of 20 Å to guide the modeling (pepATTRACT-local protocol) (*Schindler et al., 2015a*). After molecular dynamics refinement (see above), which did not apply crosslinking restraints for technical reasons, models were filtered for those that still satisfied the distance restraints provided by the crosslinks, yielding 383 models. All figures were created using PyMOL (www.pymol.org).

## Acknowledgements

We thank Geeta Narlikar (UC San Francisco, USA), Hal Alper (UT Austin, USA) and Arie Geerlof (Helmholtz Center, Munich, Germany) for plasmids and protocols, Axel Imhof (LMU, Munich, Germany) for access to MS machines, Silvia Härtel for technical help and Philipp Korber (LMU, Munich, Germany) and Peter Becker (LMU, Munich, Germany) for discussions. JL and AKS are grateful to the Schering Stiftung and the DAAD, respectively, for predoctoral fellowships. This work was funded by DFG grants MU 3613/1–1, MU 3613/3–1 and SFB 1064/1 TP-A07 awarded to FMP and the Center of Integrated Protein Science Munich (CIPSM) to MZ, respectively.

# Additional information

## Funding

| Funder | Grant reference number | Author |
|---|---|---|
| Ernst Schering Foundation | | Johanna Ludwigsen |
| Deutscher Akademischer Austauschdienst | | Ashish K Singh |
| Deutsche Forschungsgemeinschaft | CIPSM | Martin Zacharias |
| Deutsche Forschungsgemeinschaft | MU 3613/1-1 | Felix Mueller-Planitz |
| Deutsche Forschungsgemeinschaft | MU 3613/3-1 | Felix Mueller-Planitz |
| Deutsche Forschungsgemeinschaft | SFB 1064/1TP-A07 | Felix Mueller-Planitz |

The funders had no role in study design, data collection and interpretation, or the decision to submit the work for publication.

## Author contributions

JL, Conceptualization, Investigation, Visualization, Writing—original draft, Writing—review and editing; SP, Investigation, Visualization, Writing—review and editing; AKS, Investigation, Visualization, Writing—review and editing; CS, Software, Investigation, Visualization, Writing—review and editing; NH, Formal analysis, Validation, Visualization; IF, Visualization, Methodology; MZ, Conceptualization, Software, Supervision, Funding acquisition, Project administration; FM-P, Conceptualization, Software, Supervision, Funding acquisition, Investigation, Writing—original draft, Project administration, Writing—review and editing

## Author ORCIDs

Felix Mueller-Planitz, http://orcid.org/0000-0001-8273-6473

# Additional files

## Major datasets

The following dataset was generated:

| Author(s) | Year | Dataset title | Dataset URL | Database, license, and accessibility information |
|---|---|---|---|---|
| Johanna Ludwigsen, Sabrina Pfennig, Ashish K Singh, Christina Schindler, Nadine Harrer, Ignasi Forné, Martin Zacharias, Felix Mueller-Planitz | 2017 | Concerted regulation of ISWI by an autoinhibitory domain and the H4 N-terminal tail | http://www.ebi.ac.uk/pride/archive/projects/PXD005831 | Publicly available at the PRIDE Archive (accession no. PXD005831) |

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
