## [Decision Letter]

[Editors’ note: this article was originally rejected after discussions between the reviewers, but the authors were invited to resubmit after an appeal against the decision.]

Thank you for submitting your work entitled "Concerted regulation of ISWI by an autoinhibitory domain and the H4 N-terminal tail" for consideration by *eLife*. Your article has been reviewed by three peer reviewers, one of whom, Jerry L Workman (Reviewer #1), is a member of our Board of Reviewing Editors, and the evaluation has been overseen by a Senior Editor.

Our decision has been reached after consultation between the reviewers. Based on these discussions and the individual reviews below, we regret to inform you that your work will not be considered further for publication in *eLife*.

Reviewers 2 and 3 raise several technical issues with the study that could not be addressed in a timely manner. All three reviews are included below. We hope that you will find the comments in these reviews helpful in the continuation of this study.

Reviewer #1:

This is a nice biochemical study examining the auto regulation domains of the ISWI ATPase chromatin remodeling complex. The manuscript provides novel data that "correct" aspects of previous models of ISWI regulation while confirming other aspects. The most important finding in the manuscript is that the ppHSA domain of the N terminal region binds to lobe 2 of the ATPase domain to suppress the ATPase activity in the absence of the H4 tail similar to chromodomains of CHD1 binding the ATPase domain.

1) While high profile papers have been published on examining auto regulation of ISWI including some which this study corrects, the impact of this work would be greatly enhanced if it were demonstrated that the ppHSA domain in other chromatin remodelers (Ino80, Lsh, Snf2) were also found to bind and regulate their ATPases. Even one more example of this regulation would turn this detailed study of ISWI regulation into revealing novel mechanisms for a whole group of remodelers. I strongly recommend the authors try the cross linking experiment with another remodeler.

2) ISWI is complexed with other proteins. The manuscript should discuss this in the Introduction and any implications other subunits might have for the activities shown here in the Discussion.

3) The raw data for the restriction enzyme site accessibility assays should be shown somewhere in the manuscript.

Reviewer #2:

The ISWI chromatin remodeling enzymes are activated by interaction with the histone H4 N-terminal tail domain. In the current model for ISWI activation, the H4-tail binding site is blocked by the AutoN-motif in the N-terminal region (NTR) of ISWI, which contains a sequence very similar to the basic patch in the H4 tail. Consequently, it was proposed that H4 and AutoN compete for the same site on the ISWI ATPase domain. Using a combination of site-specific cross-linking, mass spectroscopy and structural modeling, the authors have mapped the approximate binding sites of the NTR and the H4 tail within purified ISWI enzymes in vitro. They deduce that the NTR and the H4 tail both bind to lobe 2 of the ATPase domain, but not at the same site. Their data are therefore inconsistent with the current model. They propose an allosteric model instead. This study also identifies two other important regions in the NTR: the ppHSA motif, which is shown to contribute to the structural stability of the NTR domain, and the AcidicN motif which, together with AutoN, is important for H4-dependent enzyme activation. Mutations in AcidicN or AutoN activate the ISWI ATPase in the absence of chromatin and allow remodeling of nucleosomal arrays even if the H4 tail is absent. These two motifs together confer H4-tail dependence on the enzyme. The authors also point out that ISWI is more similar to Chd1 than expected, given the lack of sequence homology between the NTRs of the two enzymes.

Overall, this is a thorough and careful study with interesting and important conclusions concerning the mechanism of chromatin remodeling by ISWI enzymes. The authors used both UV and chemical cross-linking to map binding sites on two different ISWI enzymes (*Drosophila* ISWI and human hSNF5) in the presence and absence of nucleosomes or H4 peptide, with consistent results. They also assessed the effects of NTR mutations in vivo, although this had to be done in yeast, with the ISW1 enzyme, which is unlike ISWI because it is inactive without Ioc subunits.

I have only one important concern. The weak point of the study is that the authors have to use structural modeling to interpret the cross-linking pattern. There is no good way around this problem in the absence of a crystal structure for a complete ISWI enzyme. The structural model in Figure 5 indicates clearly that there is no overlap between the likely binding sites on the ATPase domain for the H4 tail and the NTR domain. However, the H4 tail is apparently modeled as an α-helix, which makes it much shorter than the extended random coil structure usually proposed for the histone tails. Given that the mapped site-specific contacts involve cross-links from the first or tenth residues of the H4 tail (T1C or L10C), whereas the basic patch includes residues 16 to 20, is it possible to dock a random coil model for the H4 tail such that the mapped contacts involving T1C and L10C are maintained and the basic patch residues are located at the same site on the ATPase as AutoN, as predicted by the competition model?

Reviewer #3:

The authors analyze the functional role of three conserved motifs in the N-terminus of ISWI, an ATP-dependent chromatin remodeler that spaces nucleosomes. Previous studies had focused on what is called the AutoN domain and its sequence similarity with the histone H4 tail. Earlier data had suggested the AutoN negatively regulates the ATPase and remodeling activities of ISWI which can be competed away and relieved by the H4 tail. They present a different explanation in which the AutoN and a flanking region called AcidicN bind to a region of the ATPase domain different than the H4 tail. Instead of competing the H4 tail, AutoN and AcidicN cooperate together to regulate ISWI in an H4 tail dependent manner. A third motif called ppHSA is also presented to bind to lobe 2 of the ATPase domain and stabilizes the interactions of AutoN and AcidicN with lobe 2.

They map the interactions of the AutoN, AcidicN, ppHSA and H4 tail with the ATPase domain of ISWI using a combination of photoaffinity labeling and chemical crosslinking. They incorporate photoreactive amino acid analogs either into a site of the ATPase domain they propose to be the binding site of the NTR region or into the H4 tail. Photoaffinity labeling is a powerful technique, but there are certain concerns associated with this approach which need to be considered. Whenever a modification is made, especially one as bulky as benzophenone (essentially 2 benzene rings), it is critical to determine if the modification interferes with the normal function of the protein. The authors have not shown if the modifications in ISWI or the H4 tail peptide interfered with their binding or the associated enzymatic activity. Since their results and conclusions are heavily dependent on these crosslinking data, it is imperative for them to do the necessary controls. They need to determine if the modifications interfered with the ATPase and nucleosome remodeling activity of ISWI and if modifications in the H4 tail interfere with it stimulating the ATPase activity of ISWI with DNA. The other approach uses chemical crosslinking of proximal lysines with a reagent called BS3 and identification of crosslinked pairs by mass spectrometry. The authors failed to present the full spectral dataset of these crosslinking experiments and how they were validated or confirmed. Instead they indicate only one of the potentially many crosslinks they obtained and in this case have picked the one that is consistent with the model. The data needs to be fully presented and properly vetted before it can be accepted.

There are additional concerns about the photocrosslinking and the use of Bpa because of the type of photochemistry it entails. It is reasonably probable that the photocrosslinking captures a minor population of the protein and may not be a good representation of the bulk protein and the true state of the protein. Possibly only an inactive fraction of the protein is what is being productively crosslinked and thus additional tests need to be in place to confirm the protein interactions. The site-directed mutagenesis unfortunately do not accomplish this purpose, but only demonstrate that particular residues are important for the enzyme's activity. This is further compounded because unlike aryl azides, benzophenone has the ability to cycle between its excited state and ground state until it reaches the appropriate substrate and it has a high preference for Met. Therefore these reagents can search for quite a while until the right substrate presents itself and for these reason make it even more probable of trapping an inactive or inappropriate conformation that may occur infrequently.

A major concern of this paper is that there is a lot of in silico work done that is more speculative in terms of determining the binding surface of NTR in the ATPase domain rather than direct experimental data. Based on only one crosslinked site and additional assumptions does not provide a high confidence in the sites proposed by the authors. This non-objective way of showing the data appears multiple times in this manuscript. And yet this is a major point of the manuscript. Also when they dock the H4 tail on the surface of the ATPase domain based on crosslinking at two sites on H4, they only show one model instead of the family of possible interactions obtained by their modeling method. The choice of only one site on the surface of the ATPase domain seems to be inadequate for probing the interactions of the NTR with the ATPase domain. The rationale for choosing the site based on the premise that it should be the same or similar to that of the chromodomains of Chd1 when there is no sequence homology between these two region doesn't have much supporting evidence and is very tenuous. There should at least by other sites within lobe 1 that are examined as controls in order to confirm the specificity of the observed contacts.

The choice of protein constructs was somewhat curious and varied. The authors start with ISWI from *Drosophila* that contains most of the full length ISWI including the HSS tail, but then revert to one missing the HSS region. And then a third variation is using the human SNF2H when doing the chemical crosslinking with BS3. All of these variations are most likely going to matter. I think it is important for the authors to do all the experiments with the ISWI construct containing the HSS region, because the HSS region is very likely to matter in this regard.

The other point of the paper is the ppHSA motif stabilizes the interactions of the NTR, however this does not seem to be well supported. The authors examine recombinant constructs of ISWI with serial truncations from the N-terminus. They examined the stabilities of these proteins using differential scanning fluorimetry and concluded the deletion of the extreme N-terminus containing the ppHSA motif is less stable than the full length ISWI. The data however shows that the truncated proteins are more complex that full length ISWI and did not give a simple one curve. Indeed there seems to be components within the sample that may actually be more stable than full length. The purity of these proteins as visualized by SDS-PAGE analysis are not very high and could lead to the complicated scan of these truncated proteins. The author average all this together to give them a lower thermal stability for the truncated constructs and is misleading. The also conclude that the ppHSA motif dies not contribute to the regulation of the activity of ISWI; however they see a 3.6 fold reduction when this motif is removed. I would have thought that was a substantial effect and should not be discounted.

---

## [Author Response]

Reviewer #1:

This is a nice biochemical study examining the auto regulation domains of the ISWI ATPase chromatin remodeling complex. The manuscript provides novel data that "correct" aspects of previous models of ISWI regulation while confirming other aspects. The most important finding in the manuscript is that the ppHSA domain of the N terminal region binds to lobe 2 of the ATPase domain to suppress the ATPase activity in the absence of the H4 tail similar to chromodomains of CHD1 binding the ATPase domain.

1) While high profile papers have been published on examining auto regulation of ISWI including some which this study corrects, the impact of this work would be greatly enhanced if it were demonstrated that the ppHSA domain in other chromatin remodelers (Ino80, Lsh, Snf2) were also found to bind and regulate their ATPases. Even one more example of this regulation would turn this detailed study of ISWI regulation into revealing novel mechanisms for a whole group of remodelers. I strongly recommend the authors try the cross linking experiment with another remodeler.

We agree that additional crosslinking experiments with other remodeler families containing a ppHSA domain would be very interesting. We feel however that this is experimentally out of reach and may be beyond the scope of our study. Purifying megadalton complexes such as Ino80 and Snf2 is technically demanding, and incorporation of the photo-crosslinkable amino acids is not trivial either. A comparative analysis of the role of the ppHSA motif across multiple remodeler families -while certainly worthwhile in its own right- would possibly require a year or more for completion. In addition, it would not substantiate major conclusions of our study, which relate to the mechanism of H4 tail recognition by ISWI and the structure and function of its N-terminal region (NTR). For these reasons, we opted not to pursue crosslinking experiments with other remodelers.

2) ISWI is complexed with other proteins. The manuscript should discuss this in the Introduction and any implications other subunits might have for the activities shown here in the Discussion.

As suggested, we now include more references in the text to the presence of additional subunits in ISWI complexes. Their function is not well understood but they potentially provide a whole new layer of regulation:

Introduction section: “Cells also adjust the subunit composition of remodeling complexes during development (Lessard et al.,2007).”

Introduction section, paragraph 4: “Another layer of regulation is imposed by the non-catalytic subunit termed ACF1, which associates with ISWI and sequesters the H4 tail under certain conditions (Hwang et al.,2014).”

Discussion section: “Our data do not rule out additional binding sites on other domains and on ISWI’s partner subunit ACF1 as proposed earlier (Boyer, Latek and Peterson, 2004, Grune et al., 2003 and Hwang et al,. 2014).”

Discussion section, last paragraph: “Sensitive biophysical assays will be instrumental in the future for resolving conformational changes that may occur during H4 tail recognition and for understanding their functional importance in ISWI complexes.”

3) The raw data for the restriction enzyme site accessibility assays should be shown somewhere in the manuscript.

As requested, we incorporated the raw data (Figure 3—figure supplement 3; Figure 7—figure supplement 2; Figure 8—figure supplement 1).

Reviewer #2:

*The ISWI chromatin remodeling enzymes are activated by interaction with the histone H4 N-terminal tail domain. In the current model for ISWI activation, the H4-tail binding site is blocked by the AutoN-motif in the N-terminal region (NTR) of ISWI, which contains a sequence very similar to the basic patch in the H4 tail. Consequently, it was proposed that H4 and AutoN compete for the same site on the ISWI ATPase domain. Using a combination of site-specific cross-linking, mass spectroscopy and structural modeling, the authors have mapped the approximate binding sites of the NTR and the H4 tail within purified ISWI enzymes* in vitro. They deduce that the NTR and the H4 tail both bind to lobe 2 of the ATPase domain, but not at the same site. Their data are therefore inconsistent with the current model. They propose an allosteric model instead. This study also identifies two other important regions in the NTR: the ppHSA motif, which is shown to contribute to the structural stability of the NTR domain, and the AcidicN motif which, together with AutoN, is important for H4-dependent enzyme activation. Mutations in AcidicN or AutoN activate the ISWI ATPase in the absence of chromatin and allow remodeling of nucleosomal arrays even if the H4 tail is absent. These two motifs together confer H4-tail dependence on the enzyme. The authors also point out that ISWI is more similar to Chd1 than expected, given the lack of sequence homology between the NTRs of the two enzymes.

Overall, this is a thorough and careful study with interesting and important conclusions concerning the mechanism of chromatin remodeling by ISWI enzymes. The authors used both UV and chemical cross-linking to map binding sites on two different ISWI enzymes (Drosophila ISWI and human hSNF5) in the presence and absence of nucleosomes or H4 peptide, with consistent results. They also assessed the effects of NTR mutations in vivo, although this had to be done in yeast, with the ISW1 enzyme, which is unlike ISWI because it is inactive without Ioc subunits.

We thank the reviewer for appreciating our efforts as “a thorough and careful study with interesting and important conclusions concerning the mechanism of chromatin remodeling by ISWI enzymes”.

I have only one important concern. The weak point of the study is that the authors have to use structural modeling to interpret the cross-linking pattern. There is no good way around this problem in the absence of a crystal structure for a complete ISWI enzyme. The structural model in Figure 5 indicates clearly that there is no overlap between the likely binding sites on the ATPase domain for the H4 tail and the NTR domain. However, the H4 tail is apparently modeled as an α-helix, which makes it much shorter than the extended random coil structure usually proposed for the histone tails. Given that the mapped site-specific contacts involve cross-links from the first or tenth residues of the H4 tail (T1C or L10C), whereas the basic patch includes residues 16 to 20, is it possible to dock a random coil model for the H4 tail such that the mapped contacts involving T1C and L10C are maintained and the basic patch residues are located at the same site on the ATPase as AutoN, as predicted by the competition model?

We realize that we did not sufficiently explain the modeling procedure and we would like to clarify this point. Contrary to the reviewer’s concern, the H4 tail was not modeled as an α-helix. Indeed, the modeling algorithm was not fed with any information on the tail structure. Accordingly, the family of 383 models that fit the distance restraints provided by the crosslinks, contained both helical und ‘unstructured’ H4 tails (see also Materials and methods for details). The modeled interaction surface (red and yellow in Figure 5) depicts the contact frequency between ISWI and the H4 tail across the entire family of 383 models.

Nevertheless, the H4-tail assumed a helical organization in the top three best fitting models, one of which was depicted in former Figure 5. Such a helical configuration would indeed be in good agreement with previously published results [1-3]. We therefore showed one helical model for illustration purposes only in our previously submitted manuscript.

By doing so, we may have created the wrong impression that we assumed the tail to be α helical during modeling, or that the H4 tail must form a helix once bound to ISWI. We therefore took out the modeled H4 tail from the revised figure. Importantly, all information about the family of models (helical and extended) was and still is contained in the colored interaction surface. See also our response to reviewer #3.

Reviewer #3:

The authors analyze the functional role of three conserved motifs in the N-terminus of ISWI, an ATP-dependent chromatin remodeler that spaces nucleosomes. Previous studies had focused on what is called the AutoN domain and its sequence similarity with the histone H4 tail. Earlier data had suggested the AutoN negatively regulates the ATPase and remodeling activities of ISWI which can be competed away and relieved by the H4 tail. They present a different explanation in which the AutoN and a flanking region called AcidicN bind to a region of the ATPase domain different than the H4 tail. Instead of competing the H4 tail, AutoN and AcidicN cooperate together to regulate ISWI in an H4 tail dependent manner. A third motif called ppHSA is also presented to bind to lobe 2 of the ATPase domain and stabilizes the interactions of AutoN and AcidicN with lobe 2.

They map the interactions of the AutoN, AcidicN, ppHSA and H4 tail with the ATPase domain of ISWI using a combination of photoaffinity labeling and chemical crosslinking. They incorporate photoreactive amino acid analogs either into a site of the ATPase domain they propose to be the binding site of the NTR region or into the H4 tail. Photoaffinity labeling is a powerful technique, but there are certain concerns associated with this approach which need to be considered. Whenever a modification is made, especially one as bulky as benzophenone (essentially 2 benzene rings), it is critical to determine if the modification interferes with the normal function of the protein. The authors have not shown if the modifications in ISWI or the H4 tail peptide interfered with their binding or the associated enzymatic activity. Since their results and conclusions are heavily dependent on these crosslinking data, it is imperative for them to do the necessary controls. They need to determine if the modifications interfered with the ATPase and nucleosome remodeling activity of ISWI and if modifications in the H4 tail interfere with it stimulating the ATPase activity of ISWI with DNA.

We agree that photo-crosslinking is a powerful technique but that the functional consequences of incorporation of the benzophenone moiety into proteins need to be assessed. In the revised manuscript, we now show the requested quality controls that probe whether benzophenone incorporation affects function.

We include now a figure that shows that benzophenone-containing H4 tail peptides retain their ability to stimulate ATP hydrolysis (Figure 5—figure supplement 2), that benzophenone-labeled nucleosomes stimulate ATP hydrolysis like wild-type nucleosomes (Figure 5—figure supplement 2) and that these nucleosomes can associate with ISWI (Figure 5—figure supplement 2).

We also assessed the effect of Bpa substitution of histidine 483. ISWI_H483B_ hydrolyzed ATP and remodeled nucleosomes 3- to 4-fold more slowly than ISWI_WT_ (new Figure 4—figure supplement 1; Figure 6—figure supplement 4), a result that may not be surprising given that the mutation is located in the conserved ‘block D’ of Snf2 ATPases. Importantly, coupling of ATP hydrolysis to remodeling of nucleosomes was unaffected suggesting that the mutant remained structurally largely intact. We also probed whether auto-regulation of ISWI_H483B_ by its NTR was affected by the H483B mutation. Mutagenesis of the NTR (+3 and +6 mutation) had analogous effects on ISWI_WT_ and ISWI_H483B_ (new Figure 6—figure supplement 4), demonstrating that the function of the NTR was largely unperturbed. In conclusion, we think that our new data sufficiently justify the use of ISWI_H483B_. These new results are discussed in subsection “The NTR contacts Lobe 2 of the ATPase domain”.

The other approach uses chemical crosslinking of proximal lysines with a reagent called BS3 and identification of crosslinked pairs by mass spectrometry. The authors failed to present the full spectral dataset of these crosslinking experiments and how they were validated or confirmed. Instead they indicate only one of the potentially many crosslinks they obtained and in this case have picked the one that is consistent with the model. The data needs to be fully presented and properly vetted before it can be accepted.

We realize that the inclusion of the chemical crosslink obtained with BS3 was confusing. The reviewer correctly predicts that the chemical crosslinking dataset contained many crosslinks. Because we are currently preparing a separate manuscript about these crosslinks, we could not include the entire dataset here. Importantly, none of the additional chemical crosslinks questioned any of our conclusions, nor did they substantiate any of the claims, so we didn’t think it necessary to report them. We would be happy to share the entire dataset with the reviewer as necessary.

Actually, none of our conclusions in the manuscript relied on this chemical crosslink. Being a low confidence crosslink, we did not even use it for structural docking purposes. Because inclusion of the chemical crosslink evidently led to more question than answers, we simply removed it from the manuscript during the revision.

There are additional concerns about the photocrosslinking and the use of Bpa because of the type of photochemistry it entails. It is reasonably probable that the photocrosslinking captures a minor population of the protein and may not be a good representation of the bulk protein and the true state of the protein. Possibly only an inactive fraction of the protein is what is being productively crosslinked and thus additional tests need to be in place to confirm the protein interactions. The site-directed mutagenesis unfortunately do not accomplish this purpose, but only demonstrate that particular residues are important for the enzyme's activity. This is further compounded because unlike aryl azides, benzophenone haas the ability to cycle between its excited state and ground state until it reaches the appropriate substrate and it has a high preference for Met. Therefore these reagents can search for quite a while until the right substrate presents itself and for these reason make it even more probable of trapping an inactive or inappropriate conformation that may occur infrequently.

We thank the reviewer for this detailed comment and agree that -like any other method- also photo- crosslinking has its disadvantages. The disadvantage that the reviewer pointed out is that a very flexible structure may assume multiple conformations and crosslink only out of some of them. We better discussed this caveat in the revised manuscript:

Subsection “The H4 tail binds Lobe 2 adjacent to AutoN-AcidicN”: “Note that not all lower quality crosslinks were compatible with this binding mode, possibly because the H4-tail peptide bound flexibly or in multiple binding modes (Racki et al., 2014). Some of these modes may not be strongly populated or functionally active as crosslinking can in principle trap fleeting intermediates.”

We would like to point out that all our crosslinks, including the crosslinks formed by ISWI_H483B_, are fully supported by a crystal structure that was published during the revision of this manuscript (Yan et al., 2016). This structure therefore provided strong orthogonal validation of our crosslinking data, and we discuss our results in the light of the new structure briefly in the Discussion section now.

A major concern of this paper is that there is a lot of in silico work done that is more speculative in terms of determining the binding surface of NTR in the ATPase domain rather than direct experimental data. Based on only one crosslinked site and additional assumptions does not provide a high confidence in the sites proposed by the authors. This non-objective way of showing the data appears multiple times in this manuscript. And yet this is a major point of the manuscript.

To clarify, we performed in silico docking only twice. We docked the H4 tail, and we docked AutoN- AcidicN. Docking of the H4 tail was well constrained by the experimentally observed crosslinks between H4 and the enzyme, for which we comprehensively disclosed all supporting spectra (Figure 5; Table 3). Crosslink-guided structural modeling is the current state of the art, and many laboratories use it when higher resolution techniques cannot readily be employed. The position of the modeled H4 tail was indeed fully consistent with a crystal structure that came out during the revision of this manuscript (Yan et al., 2016)). Only the basic patch of H4 is visible in the structure, however, whereas our crosslinking data restrained amino acids 1 and 10, so that our work predicts the location of the entire binding pocket.

The reviewer is right, however, that we did not have distance restraints for the second docking that we performed, docking of AutoN-AcidicN. We therefore validated the proposed interface on Lobe 2 by mutagenesis during the revision (new Figure 7, Figure 7—figure supplement 1, Figure 7—figure supplement 2). Remarkably, interface mutants practically phenocopied AcidicN mutants, providing strong evidence for correct identification of the AcidicN binding pocket. The crystal structure (Yan et al., 2016) that became available after we had designed the mutagenesis experiment, fully supported the predicted AcidicN interface (new Figure 4—figure supplement 3).

*Also when they dock the H4 tail on the surface of the ATPase domain based on crosslinking at two sites on H4, they only show one model instead of the family of possible interactions obtained by their modeling method.*

We apologize for being unclear about how the modeling results were being displayed. We do, actually, show the results of the whole family of models in Figure 5. This information is contained in the yellow to red color scale, which denotes low to high contact probabilities between the H4 tail and Lobe 2. These contact probabilities were calculated from the top ranking 383 models that we obtained from docking (see also Materials and methods section). In addition, we previously displayed the H4 tail of the top ranking model.

To avoid giving the impression that we only show results of one model, we simply deleted the top ranking H4 tail model from the figure during the revision. Importantly, all information about the family of docked structures was and still is represented in the yellow to red interaction surface. See also our response to a comment from reviewer #2.

The choice of only one site on the surface of the ATPase domain seems to be inadequate for probing the interactions of the NTR with the ATPase domain. The rationale for choosing the site based on the premise that it should be the same or similar to that of the chromodomains of Chd1 when there is no sequence homology between these two region doesn't have much supporting evidence and is very tenuous. There should at least by other sites within lobe 1 that are examined as controls in order to confirm the specificity of the observed contacts.

This is a good point. We have done the requested specificity control already in a previous study where we inserted Bpa residues at different sites into the ISWI ATPase domain, including into Lobe 1. None of the positions formed crosslinks to the ISWI NTR. The NTR-Lobe2 interaction therefore appears to be specific to the region in the vicinity of H483. We have made the following changes to the text:

Subsection “The NTR contacts Lobe 2 of the ATPase domain”: “In our previous work, we incorporated Bpa in a variety of places on Lobes 1 and 2 of the ATPase domain but never observed crosslinks to the NTR (Forne et al., 2012 and Mueller-Planitz, 2015). We therefore suggest that the ppHSA motif specifically docked to a location in proximity of amino acid 483 in Lobe 2.”

Of note, all our crosslinks between Lobe 2 and the NTR are fully consistent with a recent structure that came out during the revision of this manuscript (Yan et al., 2016). Thus, our crosslinking data and the crystal structure mutually validate each other.

The choice of protein constructs was somewhat curious and varied. The authors start with ISWI from Drosophila that contains most of the full length ISWI including the HSS tail, but then revert to one missing the HSS region. And then a third variation is using the human SNF2H when doing the chemical crosslinking with BS3. All of these variations are most likely going to matter. I think it is important for the authors to do all the experiments with the ISWI construct containing the HSS region, because the HSS region is very likely to matter in this regard.

The reviewer makes the point that we should use ISWI that includes the HSS domain for all experimental approaches. Maybe we were a bit unclear about this, but we did. The yeast Isw1 experiments included the HSS domain, as did the NTR mutants that we created in *Drosophila* ISWI. Also the crosslinking experiments using benzophenone-labeled nucleosomes and ISWI_H483B_included the HSS domain.

In addition, we chose to support selected conclusions with independent data that were obtained with HSS-deleted ISWI (crosslinking of ISWI_△HSS_ to a BPA-containing H4 tail peptide and crosslinking of a ISWI_H483B, △HSS_ double mutant). Importantly, results from these experiments merely supported the evidence collected already with full-length ISWI so that none of our conclusions rest on results obtained with ISWI_△HSS_.

The reviewer seems to also suggest that we should focus on *Drosophila* ISWI and not include the human homolog SNF2H. Again, we included SNF2H only to support data that we obtained already for ISWI. None of the conclusions rest on the use of SNF2H. The one place where we used both ISWI and SNF2H was crosslinking to benzophenone-labeled nucleosomes. Importantly, we could map crosslinks to Lobe 2 for both enzymes. We do feel that inclusion of the SNF2H results strengthens the conclusion that the H4 tail binds Lobe 2 and therefore would like to include them in the manuscript.

The other point of the paper is the ppHSA motif stabilizes the interactions of the NTR, however this does not seem to be well supported. The authors examine recombinant constructs of ISWI with serial truncations from the N-terminus. They examined the stabilities of these proteins using differential scanning fluorimetry and concluded the deletion of the extreme N-terminus containing the ppHSA motif is less stable than the full length ISWI. The data however shows that the truncated proteins are more complex that full length ISWI and did not give a simple one curve. Indeed there seems to be components within the sample that may actually be more stable than full length. The purity of these proteins as visualized by SDS-PAGE analysis are not very high and could lead to the complicated scan of these truncated proteins. The author average all this together to give them a lower thermal stability for the truncated constructs and is misleading.

We agree and took out the supplemental figure altogether because we think that we can support structural destabilization of NTR mutants better otherwise: all NTR mutants proved exceedingly difficult to purify and we had to screen through a variety of different purification strategies for each of them, suggesting that they are misfolded and easily aggregate. The following changes to the text were made:

Subsection “The ppHSA motif does not substantially contribute to catalysis”: “Although ISWI variants carrying mutations or deletions in the NTR generally expressed well, we failed to purify them using standard protocols. For each ISWI variant, we screened through a variety of expression and purification strategies to improve the yield of soluble protein. The strategies that we employed included fusion to solubility tags (Z2, GB1, NusA, TrxA), fusion to or co-expression of chaperones (trigger factor, GroES/GroEL, DnaK/DnaJ/GrpE) and inclusion of protease sites (3C) at three locations in the NTR to cleave off parts of the N-terminus after purification. The strategies that proved successful are summarized schematically in Figure 3—figure supplement 1 and Figure 6—figure supplement 1.”

The also conclude that the ppHSA motif dies not contribute to the regulation of the activity of ISWI; however they see a 3.6 fold reduction when this motif is removed. I would have thought that was a substantial effect and should not be discounted.

We agree and revised the wording. Please note, however, the ppHSA motif relative to AcidicN seems to play at most a modest role for catalysis.

Subsection “The ppHSA motif does not substantially contribute to catalysis”: “In conclusion, ATPase and remodeling data suggested that both ppHSA and AT-hook are not absolutely required for catalysis in vitro. The modest decreases in remodeling activities could be due to lower stability of these enzymes (see above).”

**References**

1. Yang, D. and G. Arya, *Structure and binding of the H4 histone tail and the effects of lysine 16 acetylation.* Phys Chem Chem Phys, 2011. **13**(7): p. 2911-21.

2. Johnson, L.M., G. Fisher-Adams, and M. Grunstein, *Identification of a non-basic domain in the histone H4 N-terminus required for repression of the yeast silent mating loci.* EMBO J, 1992. **11**(6): p. 2201-9.

3. Hansen, J.C., C. Tse, and A.P. Wolffe, *Structure and function of the core histone N-termini: more than meets the eye.* Biochemistry, 1998. **37**(51): p. 17637-41.